# A local to national-scale inverse modeling system to assess the potential of spaceborne $CO_2$ measurements for the monitoring of anthropogenic emissions

Diego Santaren[1], Grégoire Broquet[1], François-Marie Bréon[1], Frédéric Chevallier[1], Denis Siméoni[2], Bo Zheng[1], Philippe Ciais[1]

[1]Laboratoire des Sciences du Climat et de l'Environnement, CEA-CNRS-UVSQ, Gif-sur-Yvette, France
[2]Thales Alenia Space, La Bocca, France

*Correspondence to*: Diego Santaren (diegosantaren@gmail.com)

**Abstract.** This work presents a flux inversion system for assessing the potential of new satellite imagery measurements of atmospheric $CO_2$ to monitor anthropogenic emissions at scales ranging from local intense point sources to regional and national scales. Such imagery measurements will be provided by the future Copernicus Anthropogenic Carbon Dioxide Monitoring Mission (CO2M). While the modeling framework keeps the complexity of previous studies focused on individual and large cities, this system encompasses a wide range of sources to extend the scope of the analysis. This atmospheric inversion system uses a zoomed configuration of the regional transport model CHIMERE which covers most of Western Europe with a 2-km resolution grid over northern France, western Germany and Benelux. For each day of March and May 2016, over the 6 hours before a given satellite overpass, the inversion controls separately the hourly budgets of anthropogenic emissions in this area from ~300 cities, power plants and regions. The inversion also controls hourly regional budgets of the natural fluxes. This enables the analysis of results at the local to regional scales for a wide range of sources in terms of emission budget and spatial extent while accounting for the uncertainties associated to natural fluxes and the overlapping of plumes from different sources. The potential of satellite data to monitor $CO_2$ fluxes is quantified with posterior uncertainties or uncertainty reductions (URs) from prior inventory-based statistical knowledge.

A first analysis focuses on the hourly to 6-hour budgets of the emissions of the Paris urban area, and on the sensitivity of the results to different characteristics of the images of vertically integrated $CO_2$ ($XCO_2$) corresponding to the spaceborne instrument: the pixel spatial resolution, the precision of the $XCO_2$ retrievals per pixel, and the swath width. This sensitivity analysis provides a correspondence between these parameters and thresholds on the targeted precisions of emission estimates. However, the results indicate a large sensitivity to the wind speed and to the *prior* flux uncertainties. The analysis is then extended to the large ensemble of point sources, cities and regions in the study domain, with a focus on the inversion system ability to monitor separately neighbor sources whose atmospheric signatures overlap and are also mixed with those produced by natural fluxes. Results highlight the strong dependence of uncertainty reductions to the emission budgets, to the wind speed and on whether the focus is on point or area sources. With the system hypothesis that the atmospheric transport is perfectly known, the results indicate that the atmospheric signal overlap is not a critical issue. All the tests are conducted

considering clear-sky conditions and the limitations from cloud cover are ignored. Furthermore, in these tests, the inversion system is perfectly informed about the statistical properties of the various sources of errors that are accounted for and systematic errors in the $XCO_2$ retrievals are ignored so that the scores of URs are assumed to be optimistic. For the emissions within the 6 hours before a satellite overpass, URs of more than 50% can only be achieved for power plants and cities whose annual emissions are more than ~2 MtC·yr$^{-1}$. For regional budgets encompassing more diffuse emissions, this threshold increases up to ~10 MtC·yr$^{-1}$. The results suggest therefore an imbalance of the monitoring capabilities of the satellite $XCO_2$ spectro-imagery towards high and dense sources.

## 1. Introduction

Comprehensive information about anthropogenic $CO_2$ emissions integrated at the scale of power plants, cities, regions and countries up to the globe would allow decision makers to track the effectiveness of emission reduction policies in the context of the Paris Agreement on Climate and other voluntary emission reduction efforts. By observing the $CO_2$ plumes downwind of large cities and industrial plants, and atmospheric signals at a few to several hundred km scales, future high-resolution spectro-imagery of the column-average $CO_2$ dry air mole fraction ($XCO_2$) from space may help addressing this need (Ciais et al., 2015; Pillai et al., 2016; Pinty et al., 2017; Schwandner et al., 2017; Broquet et al., 2018). The Copernicus Anthropogenic Carbon Dioxide Monitoring mission (CO2M, (Pinty et al., 2017)) is a prominent example of such a strategy. The CO2M concept relies on a constellation of sun-synchronous satellites with $XCO_2$ spectral-imagers to be deployed from 2025 by the European Commission and the European Space Agency (ESA). It will be based on passive radiance measurements in the Short-Wave InfraRed (SWIR), a part of the spectrum that is sensitive to $CO_2$ and $CH_4$ concentrations throughout the troposphere including the boundary layer, like almost all the space missions that have been dedicated to Greenhouse Gas (GHG) monitoring so far (Crisp, 2018).

Much remains to be understood and to be developed in order to ensure that such a constellation informs about emissions with enough detail to be relevant for policy makers. In this context, Observing System Simulation Experiments (OSSEs) of atmospheric inversions with synthetic images of $XCO_2$ data have supported the design of the space missions that will monitor the anthropogenic emissions (Buchwitz et al., 2013b; Pillai et al., 2016; Broquet et al., 2018). So far, they have mainly focused on plume inversions for some large plants and cities. However, Wang et al. (2019) estimated that cities and plants emitting more than 10 MtC·yr$^{-1}$ like Berlin (in the study by Pillai et al. (2016)) and Paris (in the study by Broquet et al. (2018)) represent less than ~7% of the global $CO_2$ emissions. Furthermore, the studied cases are generally quite isolated from other large $CO_2$ sources, facilitating the distinction of their plumes in the $XCO_2$ images, while plumes from neighbor sources could overlap and hamper the attribution to the targeted city or plant. Finally, the signature of emissions in spaceborne imagery does not consist only of clear plumes from cities, industrial clusters and point sources. Despite the large atmospheric signature of the natural fluxes, atmospheric inversions may have the potential to exploit other spatial variations in $XCO_2$ fields to quantify regional to national budgets of more diffuse sources or of all types of sources, even when the

overlapping of several plumes prevent from quantifying the emissions from individual cities and point sources. Therefore, there is a need to extend the OSSEs to a representative range of sources with various emission budgets and spreads, and various distances to other major sources, and to a larger range of spatial scales.

We have developed a high-resolution inversion system for the monitoring of $CO_2$ anthropogenic emissions at spatial scales ranging from local intense point sources like industrial sites to regional and national scales. Furthermore, this system accounts for the uncertainty in the natural fluxes. Our current simulation domain covers most of Western Europe with an extensive ensemble of cities, plants and diffuse $CO_2$ emissions. We use an analytical inversion methodology, which is the most adapted approach to efficiently test a large number of observation scenarios (section 2.1.2.), a high-resolution atmospheric transport model (section 2.1.1) and a high spatial resolution distribution of the emissions derived from different inventory products developed by the Institut für Energiewirtschaft und Rationelle energieanwendung (IER) of the University of Stuttgart (section 2.2).

The analytical inversion system follows the traditional Bayesian formalism of the atmospheric inversion. Of direct relevance here, it derives uncertainty statistics of its "posterior" emission estimates for the controlled sources (plants, cities, countryside areas or whole regions) from: i) the assumed uncertainties in the budgets derived from "prior" emission inventories (built on statistics of the fossil fuel consumption, activity data and emission factors), ii) the spaceborne $XCO_2$ observation sampling and precision, and iii) its atmospheric transport model. The improvement of the knowledge on the emissions enabled by the satellite imagery is quantified here in terms of "uncertainty reduction", i.e. of the relative difference between the prior and posterior uncertainties.

The inversion system solves for hourly budgets of the emissions from the different types of sources over the 6 hours before the satellite observation of the corresponding area. Indeed, Broquet et al. (2018) showed that, due to atmospheric diffusion, the atmospheric signatures of emissions from a mega-city like Paris that are detectable in satellite $XCO_2$ images (made with current measurement capabilities) correspond approximately to the city emissions occurring within less than 6 hours before the satellite overpass. This duration should be even shorter for the range of sources analyzed in our study since most of them have lower emissions than Paris. The analysis of the results will primarily focus on the 6h-budgets of the emissions before the satellite observation. However, controlling the hourly budgets allows evaluating the capability to solve for the temporal profiles of the emissions. It also allows accounting for some level of independence between uncertainties in the emissions from different hours, which limits the ability to cross and extrapolate information throughout the 6-hour windows. This point is critical for cities whose detectable atmospheric signatures are representative of emissions on durations shorter than 6 hours, and thus for which there is no direct constraint from the satellite observation on the first hours of such 6-hour windows.

The OSSEs presented in this study use a rather simple simulation of the $XCO_2$ observation sampling and errors from a single helio-synchronous satellite over the area of interest. The aim is indeed to provide a general understanding of the performance of the inversion system and of its potential to monitor anthropogenic emissions with spaceborne $XCO_2$ imagery rather than to evaluate a precise mission configuration with precise orbital parameters and instrumental specifications. In terms of errors

in the $XCO_2$ data, the analysis focuses on random errors due to the instrumental noise that have no spatial correlations (even though the topic is explored in Appendix A). Nevertheless, a large range of values for the precision (assumed to be homogeneous in the satellite field of view), horizontal resolution and swath of the spaceborne instrument are tested to assess the impact of these parameters on the inversion results, which can potentially support the design of future missions (section 3.2.).

Furthermore, in order to get a wide range of atmospheric transport conditions (in particular of average wind speeds, (Broquet et al., 2018)) and natural flux conditions (Pillai et al., 2016), inversions are performed for each day of March and May 2016. We work with full images of the plumes from the targeted sources each day, by flying a satellite any day in their vicinity with a large swath. Partial images of the plumes will be analyzed when studying the sensitivity to the swath width (over Paris) only. The cloud cover and large aerosol loads, and the corresponding gaps in the spaceborne passive $XCO_2$ sampling are ignored. In any case, the satellite crosses the area of interest at 11:00 (local time used hereafter) similar to what is currently recommended for the CO2M mission (Pinty et al., 2017), so that the inversion controls the hourly emissions of the sources between 5:00-11:00.

The inversion system and the corresponding transport model extend from southern France to northern Germany and from western UK to eastern Germany (Fig. 1). However, the grid of the transport model is zoomed and the analysis focuses on a 2 km-resolution sub-domain covering the north of France (in particular Paris), south east England (in particular London), west Germany, Belgium, Luxembourg and the Netherlands.

The first part of the analysis concerns the monitoring of the emissions of Paris and its suburb, which represent the most populated and densest urban area of Europe. Broquet et al. (2018) chose this megacity as a study case because its emissions are high ($\sim$11-14 $MtC \cdot yr^{-1}$ for 2013 according to the AIRPARIF inventory (Staufer et al., 2016; AIRPARIF 2013)), concentrated and relatively distant from other major sources. Moreover, the topography of the region is relatively flat and the average wind speed is moderate: 7 $m \cdot s^{-1}$ at 100 m above the ground level (Broquet et al., 2018). The $XCO_2$ plume generated by the Paris emissions has a relatively simple structure that often emerges well from the background. Therefore, the monitoring of the emissions of Paris constitutes a very favorable case with respect to other cities in Europe. Broquet et al. (2018) performed some analysis of the sensitivities of the inversion results to the wind speed and to the $XCO_2$ spaceborne spectro-imagery average precision and horizontal resolution. However, they tested a limited number of values for these observation parameters, and in particular few high precisions (< 2 ppm) and a single high spatial resolution value (< 4x4 $km^2$), while the refinement of the specification of new missions requires understanding of the sensitivity to choices of precision at the 0.1 ppm scale and of resolution at the 1 $km^2$ scale. Indeed, these choices have large impacts on the design of the instrument and therefore on its cost (Pinty et al., 2017). Furthermore, Broquet et al. (2018) performed all their OSSEs with an unique hypothesis on the prior uncertainties in the 1-hour to 6-hour budgets of the emissions from Paris, while acknowledging that the specification of these prior uncertainties could have a significant impact on the results and that the uncertainties in the inventories at such a temporal scale are difficult to assess. Therefore, this study performs a deeper

investigation of the sensitivity to the observation precision and spatial resolution, to the wind speed and to the characterization of the uncertainties in the prior estimate of the emissions (section 3.2).

The second part of this study considers the full ensemble of sources, from point sources to regions, in the 2-km resolution sub-domain. This sub-domain encompasses the Netherlands, Belgium and western Germany which are characterized by densely populated areas distributed over a network of medium-sized towns and by a large number of strong point sources, with, for example, some power plants in western Germany emitting more than 5 MtC·yr$^{-1}$. The ability of the inversion system to disentangle the plumes from neighbor sources is therefore well challenged in these areas.

This paper is organized as follows: section 2 details the theoretical and practical framework of both the inversion system and the OSSEs conducted in this study. Section 3 analyses the results relative to the monitoring of the emissions of Paris and in particular the corresponding sensitivity tests (section 3.2). This section also diagnoses the potential of the inversion system to monitor the anthropogenic emissions at the point source, city and regional scales in the area where flux and concentrations are simulated at 2-km resolution (section 3.3). Section 4 addresses the robustness and extent of the conclusions that can be derived from this study and proposes some perspectives regarding the analytical inversion system and the monitoring of anthropogenic emissions based on satellite data.

## 2. Inverse modeling system and OSSEs

In the following sections, we describe the different structural elements of the analytical inversion system: the gridded inventories used to define the point or area sources to be controlled, and to map their emissions (section 2.2), the simulation of the atmospheric $CO_2$ and $XCO_2$ signatures of the controlled sources using the atmospheric transport model CHIMERE and the matrix computation of the posterior uncertainties in the emissions (sections 2.1.1 and 2.3). We also describe the observations and parameters chosen for the OSSEs in this study: the $XCO_2$ observation sampling and errors (section 2.1.2) and the prior uncertainties in the emissions, natural fluxes and boundary conditions (section 2.4). New OSSEs with the analytical inversion system can easily be conducted with other options for these observations and parameters. However, for the sake of clarity, the descriptions of i) the components of the inversion system and of ii) the options for the OSSEs are intertwined.

### 2.1. XCO$_2$: transport model simulations and pseudo-data

### 2.1.1. Simulations of CO$_2$ and XCO$_2$ with the CHIMERE model

To compute the 4D $CO_2$ signatures of surface $CO_2$ fluxes in the study domain and for March and May 2016, and of the domain $CO_2$ boundary conditions, we use the regional atmospheric transport model CHIMERE (Menut et al., 2013). This Eulerian mesoscale model was designed to simulate pollution (Pison et al., 2007) but has also been used for $CO_2$ atmospheric inversions, and in particular for city-scale inversions of the emissions from Paris (Bréon et al., 2015; Staufer et

al., 2016; Broquet et al., 2018). It has shown high skill in simulating the daily and synoptic variability of the atmospheric $CO_2$ concentrations at European $CO_2$ continuous measurement sites (Patra et al., 2008).

The domain of our CHIMERE configuration covers most of Western Europe (Fig. 1) between latitudes ~42°N (northern Spain) and ~56°N (northern Germany) and between longitudes 6°W (eastern Ireland) and ~17°E (eastern Germany). The horizontal resolution of the zoomed grid of this configuration ranges from 2 to 50 km; the 2 km × 2 km resolution sub-domain being appropriate to simulate the atmospheric signature of a dense network of sources in northern France, Belgium, the Netherlands, Luxembourg and western Germany. The zoom and extent of the CHIMERE grid together link the simulation of $CO_2$ at local scales in this area of interest with the transport of $CO_2$ at the European scale while mitigating the computational cost. The model has 29 sigma vertical layers that extend from the surface to 300 hPa. Model concentration outputs are averaged at the hourly scale. The meteorological forcing is from the 9 km × 9 km- and 3-hour-resolution analysis of the European Center for Medium-Range Weather Forecasts (ECMWF). These three-hourly fields are interpolated at the spatial and temporal resolution of CHIMERE. The $CO_2$ concentrations used to impose the conditions at the initial time and at the lateral and top boundaries of the CHIMERE domain are from the analysis of the Copernicus Atmosphere Monitoring System (CAMS, (Inness et al., 2019)) at ~16 km resolution. The products used to impose surface $CO_2$ fluxes in the model are detailed below in section 2.2.

$XCO_2$ observations and the corresponding signatures of fluxes are simulated from the $CO_2$ 3D fields from CHIMERE at 11:00. For the sake of simplicity in the OSSEs conducted here, since we use synthetic data only and a rather simple modeling of the spaceborne observation, the computation of $XCO_2$ assumes that the vertical weighting function of the $CO_2$ column-averaging (kernel) is vertically uniform. For a given model pixel at latitude *lat* and longitude *lon*, $XCO_2$ is thus computed from the vertical average of the $CO_2$ mole fractions simulated by the model:

$$XCO_2(lat, lon) = \frac{\int_{Ptop}^{Psurf} CO_2(lat,lon,P)dP + \overline{CO_2}(P_{top})P_{top}}{P_{surf(lat,lon)}},$$ (1)

where $P$ designates the atmospheric pressure, $P_{surf}$ the atmospheric surface pressure and $P_{top}$ (300hPa) the pressure at the top boundary of the model. For pressures lower than $P_{top}$, we assume that the $CO_2$ concentrations equal the horizontal average of the top-level mixing ratios in CHIMERE ($\overline{CO_2}(P_{top})$). Indeed, we do not expect significant spatial gradients of $CO_2$ over the simulation domain in the upper atmosphere. This is supported by the lack of signal in our simulations of the atmospheric signatures of the surface fluxes in the upper layer of the model.

### 2.1.2. $XCO_2$ pseudo-data sampling and error

As detailed in section 2.3 below, the OSSE framework of the inversions requires the location and time of the individual $XCO_2$ data, and the associated error statistics, but not the explicit values of the synthetic observations themselves. In this study, we consider pseudo satellites with a Low Earth Orbit (LEO) whose altitude and inclination parameters are similar to the ones of the A-Train (705 km and 98.2° respectively, (Parkinson et al., 2006)). The satellite observations are assumed to

occur at 11:00 in the morning. Successive tracks of a single satellite on this orbit are distant by ~25 degrees. However, we do neither study the potential of a specific satellite nor that of a constellation of such LEO satellites depending on their number. Furthermore, this study focuses on results at the scale of 6 hours. Therefore, it considers single satellite tracks for any day that do not correspond to a specific positioning of a satellite on the chosen orbit: when studying the emissions of the Paris urban area, we use the track that is nearly centered on this city every day and various swaths are considered to study the sensitivity of the results to this parameter (section 3.2.2). For the study of results for the ensemble of sources contained within the 2-km sub-domain (section 3.3), we use a track centered over Belgium every day and a 900-km swath to ensure a full coverage of the plumes from these sources (the sensitivity to a realistic range of swath widths is not investigated in this second set of analysis).

Our OSSEs assess the impact on the inversion results of the measurement noise from the satellite instrument only, ignoring the errors associated to the radiative transfer inverse modeling for the retrieval of the $XCO_2$ data from the radiance measurements (Buchwitz et al., 2013a; Broquet et al., 2018), and in particular any "systematic error". The errors on the $XCO_2$ data at the spatial resolution of the measurements are thus assumed to be Gaussian, unbiased and uncorrelated in space or time. The distribution of the standard deviation (STD) for these errors is also assumed to be uniform and these errors are therefore summarized by a single value of STD (denoted the data precision hereafter).

A large number of scenarios are tested for the observation specifications: the precision on the individual $XCO_2$ data varies between 0.3 and 2 ppm, and the spatial resolution of the ground pixels can take the following values: 2 km×2 km (longitude × latitude), 2 km×3 km , 3 km×3 km , 3 km×4 km and 4 km×4 km. The reference is a precision of 0.6 ppm and a spatial resolution of 2×2 km$^2$. These values are similar to the characteristics of the simulation of CO2M data used in the study of Wang et al. (2020). When studying the sensitivity of the results over Paris to the swath of the instrument, the swath is varied from 100 km to 600 km, with a reference value of 300 km.

## 2.2. CO$_2$ fluxes

### 2.2.1. Maps and time-series of anthropogenic emissions and natural fluxes

High resolution maps of anthropogenic emissions are needed to define appropriate point and areas sources to be controlled by the inversion. High resolution maps of anthropogenic and biogenic fluxes are also needed to distribute the controlled local to regional budgets of these fluxes on the spatial grid of the CHIMERE model. Finally, such maps are needed to provide insights into the typical budgets of fluxes at the control resolution, and thus to quantify the prior uncertainty in these budgets with a suitable order of magnitude.

The anthropogenic CO$_2$ emissions are extracted from several datasets compiled by IER (Pregger et al., 2007; Thiruchittampalam et al., 2012). These datasets provide maps of the annual budgets per sectors of anthropogenic activities over different domains, and at different spatial resolutions. We have merged and re-gridded them to derive a map of the annual budgets of emissions over the entire grid of the CHIMERE configuration. The emissions corresponding to France and

Germany are extracted from the respective IER national maps for 2005 at a 1-minute resolution, while emissions in Belgium, Luxembourg and the Netherlands are derived from an IER 1-km product covering northern Europe for 2005. The IER 5-km resolution map covering the whole Europe for 2008 is used for the emissions over the rest of the domain. The gridded spread sources in the IER maps are interpolated on the CHIMERE grid but the large point sources are relocated as point sources in individual CHIMERE grid cells. We then derive the hourly emission maps from the annual emission map by applying the convolution of IER typical temporal profiles specific to each country and sector. These profiles include seasonal, daily and diurnal variations of emissions for large sectors such as traffic, power demand, domestic heating or air conditioning (Pregger et al. 2007).

The IER maps for France, Germany, northern Europe and the whole Europe correspond to annual budgets for years (2005 and 2008) that can be different from one area to the other, and which are different from the year chosen for the atmospheric transport and for the natural fluxes (2016). This could raise some inconsistencies if assimilating real data in the inversion. However, this study is based on OSSEs with some strong simplifications regarding the observation system since the overarching target is a general understanding of the behavior and potential of the inversion. This requires the use of a high resolution and realistic distribution of the emissions in space and time, but not a precise estimate of their amplitudes for a given year.

The land surface natural fluxes are derived from 8-km resolution simulations made with the Vegetation Photosynthesis and Respiration Model (VPRM) model for the year 2016. This prognostic model delivers hourly values of Net Ecosystem Exchange (NEE) by assimilating satellite and meteorological data (Mahadevan et al., 2008). These values of NEE are interpolated over the CHIMERE area at the hourly time scale. Natural ocean fluxes are ignored.

### 2.2.2. Controlled areas

The resulting hourly maps of anthropogenic $CO_2$ emissions for spread sources and large point sources are decomposed spatially to define the areas for which hourly emission budgets are controlled by the inversion: large point sources, cities, remaining parts of regions from which point sources and cities have been extracted (covering diffuse emissions only), and full regions when point sources and cities are not controlled separately but altogether with the diffuse emissions. Hourly budgets of the natural fluxes are controlled for full regions only, the regions used for the control of anthropogenic and biogenic fluxes being identical.

The definition of the regions is done considering the whole domain. It corresponds to administrative regions of France, Belgium, the Netherlands, Luxembourg and Germany, and to three additional large "regions": the United Kingdom, Switzerland and the rest of the domain. This subdivision results in 67 regions (Fig.1). These 67 regions correspond to the spatial resolution of the natural fluxes in the inversion.

Point sources and cities are controlled individually in the 2-km resolution part of the CHIMERE grid only (section 2.1.1., Fig. 1). In the 39 regions entirely comprised within this sub-domain, we individually control the 84 point sources (e.g., factories, power plants...) whose annual emissions are larger than 0.2 MtonC.yr[-1]. The maps of the remaining emissions

(excluding these point sources) in each of these 39 regions are then processed to extract large urban areas to be controlled independently, ensuring at least one controlled urban area per region, and that no controlled urban area overlaps two different regions. An algorithm of pattern recognition has been designed for such an extraction, with the idea that urban areas correspond to clusters of adjacent high emitting pixels (also followed by Wang et al. (2019)). After having applied a Gaussian filter to smooth the spatial distribution of the emissions, the large urban areas are defined by a label-connecting algorithm (Stockman et al., 2001) which identifies the clusters of adjacent points whose emissions are above a predefined threshold. As the density and extension of cities vary considerably amongst the different regions, the parameters of the pattern recognition algorithm, i.e. the standard deviation of the Gaussian filtering and the emission threshold, are different for each region to ensure that each region contains at least one controlled urban area (Fig. 2). As a result, we identify 152 control urban areas within the 2-km resolution sub-domain. They are characterized by a wide range of budgets and spatial spread of their emissions, the annual budgets ranging between ~0.07 MtonC.yr$^{-1}$ and ~9.9 MtonC.yr$^{-1}$ (with a mean and a standard deviation of ~ 0.8 MtonC.yr$^{-1}$ and ~1.5 MtonC.yr$^{-1}$ respectively), and areas ranging from ~8 km$^2$ to ~2400 km$^2$ (with a median value of ~240 km$^2$).

The remaining emissions, after having extracted the large point sources and urban areas in the 39 regions, are considered to be diffuse and called hereafter the "countryside" emissions. The inversion controls their budgets in each region. The analysis of the results at the regional scale for these 39 regions will consider either the countryside emissions only (i.e. focusing on the individual control variables), or their aggregation with the emissions from the point sources and cities within the same region (i.e. considering the full geographical extent of the regions). The inversion also directly controls the total budget of the emissions for the 28 regions that are not fully comprised in the 2-km sub-domain (most of these 28 regions do not overlap this sub-domain at all). Overall, the control of countryside or total regional emissions adds 67 controlled areas (corresponding to the 67 regions) for the anthropogenic emissions so that the inversion controls the hourly budgets of anthropogenic emissions for 303 areas (84 point sources, 152 urban areas and 67 countryside or regional areas) and the hourly budgets of natural fluxes for 67 areas.

## 2.3. Analytical flux inversion

### 2.3.1 Theoretical framework

The inversion system follows a traditional analytical inversion approach based on the Bayesian formalism and assuming that error statistics follow Gaussian distributions (Tarantola et al., 1987; Broquet et al., 2018). The system controls factors that scale the hourly budgets of the different control areas for the anthropogenic and biogenic fluxes defined in section 2.2.2. It also controls a single scaling factor applied to the $CO_2$ field used to impose the initial, lateral and top $CO_2$ boundary conditions of the model, since such boundary conditions generally bear important large-scale uncertainties that can impact the estimates of sources within the domain (Broquet et al., 2018). In the OSSEs of this study, the inversion periods for each day in March or May 2016 cover the 6 hours (5:00-11:00) before 11:00, when the satellite observations are supposed to be

made. The number of control variables (2221 = 370 controlled areas × 6 time slots + 1 control variable for the boundary conditions), is sufficiently small to solve the inverse problem analytically. However, building the matrix $\mathbf{H}$ that encompasses the atmospheric transport operator and that is described below (section 2.3.2) requires a large computational burden.

For a given inversion period, we define the control vector $\mathbf{x}$ as the set of controlled scaling factors for the hourly flux budgets and the boundary conditions. The prior uncertainty in $\mathbf{x}$ is assumed to follow a Gaussian distribution and to be unbiased. It is thus characterized by the uncertainty covariance matrix $\mathbf{B}$.

In this study, the observation vector $\mathbf{y}$ is defined by the $XCO_2$ concentrations in the transport model horizontal grid cells sampled by the observations. The simulation of $\mathbf{y}$ based on a given estimate of $\mathbf{x}$ is given by the linear observation operator $\mathbf{H}$: $\mathbf{x} \rightarrow \mathbf{y} = \mathbf{Hx}$, which chains three operators. The first operator $\mathbf{H}_{\text{distr}}$ distributes the controlled hourly budgets of emissions in space within the controlled areas, and provides the spatial and temporal mapping of the boundary conditions whose scaling factor is controlled by the inversion. The second operator $\mathbf{H}_{\text{transp}}$ is the atmospheric transport from the emissions and the boundary conditions to the full $CO_2$ and $XCO_2$ fields. Finally, the third operator $\mathbf{H}_{\text{sample}}$ performs the $XCO_2$ sampling at the location of the $XCO_2$ data (section 2.1.2). Differences between $\mathbf{Hx}$ and observed values for $\mathbf{y}$ arise due, on the one hand, to uncertainties in $\mathbf{x}$, and, on the other hand, to the combination of errors in the observation operator and in the observation data that are called altogether "observation errors". The errors from the observation operator are strongly associated with the atmospheric transport model errors (Houweling et al., 2010; Chevallier et al., 2010), but also with the discretization and spatial resolution of the transport and inversion problems, which raise representation and aggregation errors (Kaminski et al., 2001; Bocquet et al., 2011). Assuming that they follow Gaussian and unbiased distributions like the prior uncertainties, these observation errors are fully characterized by the observation error covariance matrix $\mathbf{R}$. The $\mathbf{H, B}$ and $\mathbf{R}$ matrices must be explicitly estimated in the analytical inversion framework (section 2.3.2 and 2.4).

The Bayesian theory (Tarantola et al., 1987) states that the statistics of the knowledge on $\mathbf{x}$ knowing i) the prior estimate of $\mathbf{x}$, ii) the observed values for $\mathbf{y}$ and iii) $\mathbf{H}$ as a link between the $\mathbf{x}$ and $\mathbf{y}$ spaces, follow a Gaussian and unbiased distribution. The uncertainty in such a posterior estimate is thus fully characterized by the posterior uncertainty covariance matrix $\mathbf{A}$ given by:

$$\mathbf{A} = [\mathbf{B}^{-1} + \mathbf{H}^{\mathrm{T}}\mathbf{R}^{-1}\mathbf{H}]^{-1} \tag{2}$$

The analysis of $\mathbf{A}$ and its comparison to $\mathbf{B}$, aggregated or not over different spatial and temporal scales, are the critical diagnostics in this study to assess the potential of inversions assimilating $XCO_2$ images. The score of uncertainty reduction for a given flux budget is a common indicator for evaluating the performance of an observation system. It is defined as the relative difference between the STD of the prior ($\sigma_{prior}$) and posterior ($\sigma_{post}$) uncertainties in this flux budget ($UR = 1 - \frac{\sigma_{post}}{\sigma_{prior}}$). If the assimilation of satellite observations perfectly constrains a given flux budget, the corresponding UR equals 100%. If this assimilation does not provide any information on the flux budget, UR equals 0%.

## 2.3.2. Building the observation operator matrix H

The analytical inversion system is essentially built on the explicit computation of $\mathbf{H}=\mathbf{H}_{distr}\mathbf{H}_{transp}\mathbf{H}_{sample}$. The different columns of $\mathbf{H}$ correspond to the signatures (or "response functions") in the observation space of the different control variables, *i.e.* of the different hourly emissions for each control area, and of the boundary conditions. They are computed by applying the sequence of operators $\mathbf{H}_{distr}$, $\mathbf{H}_{transp}$ and then $\mathbf{H}_{sample}$ to each control variable set to 1, keeping the others null (Broquet et al., 2018). $\mathbf{H}_{distr}$ is defined based on the flux maps detailed in section 2. $\mathbf{H}_{transp}$ corresponds to the CHIMERE model and to the vertical integration of $CO_2$ into $XCO_2$ presented in section 2.1.1, while $\mathbf{H}_{sample}$ corresponds to the sampling, on the transport model grid, of the simulated $XCO_2$ values according to the spatial distribution of the pseudo-observations (section 2.1.2). A generalized $\mathbf{H}$ is actually stored for the analytical inversion system to anticipate any option for $\mathbf{H}_{sample}$, by recording the full $CO_2$ and $XCO_2$ fields from the application of $\mathbf{H}_{distr}\mathbf{H}_{transp}$ to each control variable, i.e. the full $CO_2$ and $XCO_2$ signatures of each control variable.

## 2.4. Practical implementation of the OSSEs

While, in principle, $\mathbf{R}$ should characterize both the errors in $XCO_2$ data and the errors from the observation operator $\mathbf{H}$, this study focuses on the impact of the observation sampling and errors only. It ignores the errors from the observation operator. Moreover, the observation errors are restricted to the measurement noise which is uncorrelated in space and time as detailed in section 2.1.2. The different $\mathbf{R}$ matrices used for the OSSEs (depending on the observation sampling and noise) are thus all diagonal. The errors on the individual pseudo-observations are described by an uniform precision ($\sigma_{XCO2}$) depending on the chosen satellite configuration (section 2.1.2). However, the observation vector $\mathbf{y}$ is defined by the transport model grid rather than by the precise location and coverage of the data. Therefore, the diagonal elements of $\mathbf{R}$ follow the aggregation of $n_{obs}$ pseudo observations with uncorrelated errors (where $n_{obs}$ is potentially greater than 1) within each model grid cell corresponding to an element of $\mathbf{y}$, so that the resulting STD of the errors for this element is given by $\frac{\sigma_{XCO2}}{\sqrt{n_{obs}}}$.

Prior estimates of anthropogenic emissions and biogenic fluxes are generally provided by inventories and ecosystem model simulations such as those used here in $\mathbf{H}_{distr}$ to distribute the fluxes at high resolution. $\mathbf{B}$ should characterize uncertainties in such products and is thus set with values corresponding to typical relative uncertainties in the budgets from the maps detailed in section 2.2.1. Prior estimates of the boundary conditions for regional inversions are usually interpolated from large scale analysis or inversions. Such products can bear significant large-scale errors at the boundaries of Europe (Monteil et al., 2019). We reflect it by setting in $\mathbf{B}$ the standard deviation of the prior uncertainty in the scaling factor for the boundary conditions (see below). When constructing the $\mathbf{B}$ matrices in all our OSSEs, we assume that there is no correlation between the prior uncertainties associated to different controlled emission areas or between these uncertainties and the one associated to the boundary conditions. The spatial correlations of the uncertainties in anthropogenic emission inventories is a complex topic and the current lack of characterization for such correlations led to such a conservative set-up (Wang et al. 2018; Super et al., 2020). However, we model the temporal correlations between prior uncertainties in scaling factors associated to

different hourly natural or anthropogenic flux budgets of the same controlled emission area by using an exponentially decaying function with a correlation time scale τ (like, for instance, (Bréon et al., 2015)):

$$\rho_{i,j} = e^{-\frac{|j-i|}{\tau}} \tag{3}$$

where j and i are the indices of two corresponding hours. The STD of the prior uncertainties in the scaling factors for the different hourly budgets of the same controlled area are fixed to an identical value $\sigma_{hour}$. Finally, we assume that the STD of the prior uncertainties in scaling factors for the 6-hour budgets of the natural fluxes or of the anthropogenic emissions from a controlled area during 5:00-11:00 (to be applied to the budgets from the IER and VPRM products presented in section 2.2.1 and used to build $\mathbf{H}_{distr}$) is fixed to a value $\sigma_{Budget}$ that is the same for all control areas: typically 50% or 100% of the 6-hour

budgets. The STD of the prior uncertainties in the scaling factors for the hourly budgets of the controlled areas $\sigma_{hour}$ are then derived based on these different assumptions. The reference parameters for $\mathbf{B}$ are fixed to τ = 3 hours and $\sigma_{Budget}$ = 50%. Despite the differences between the temporal variations of the hourly emissions from one control area to the other, or between natural and anthropogenic fluxes in $\mathbf{H}_{distr}$, these STD show a negligible variation of less than 1% and when considering the reference set-up for $\mathbf{B}$, $\sigma_{hour}$~65%. The sensitivity of the inversion to the values of $\sigma_{Budget}$ and τ is assessed in

section 3.2.3. Finally, we use 1% for the STD of the prior uncertainty in the scaling factor associated to the boundary conditions (i.e. typically an uncertainty of ~4 ppm in the average boundary conditions). This value is quite pessimistic, but some tests in which this value was varied (not shown) demonstrate a very weak sensitivity of the results for the fluxes to this parameter.

## 3. Results

### 3.1. High-resolution simulations of $XCO_2$

Previous sections documented how, for each 6-hour period, the inversion system exploits the simulated $XCO_2$ fields at 11:00 to constrain each hourly budget of the anthropogenic or natural fluxes of the controlled areas between 5:00 and 11:00. The CHIMERE full $XCO_2$ simulations between 5:00 and 11:00 with the anthropogenic emissions, natural fluxes and/or domain boundary conditions detailed in sections 2.1 and 2.2 are used in this section to compare the overall signatures of these

375 components and of the controlled areas, and to discuss their overlapping. Figure 2b shows the $XCO_2$ signatures of all the anthropogenic emissions in the domain except those from the 84 point sources controlled individually by the inversion in the 2-km resolution subdomain (that are illustrated in Fig. 2a). Figure 2c integrates the $XCO_2$ produced by the 84 point sources and shows the signature of all the anthropogenic emissions. Finally, figure 2d displays the superposition of the $XCO_2$ signatures of all the anthropogenic emissions, of the natural fluxes and of the boundary conditions. For all these figures,

$XCO_2$ values are taken at 11:00 and are provided by simulations between 5:00 and 11:00 on May 23rd which is a day of strong northwest wind (~10 m·s-1 over Paris at 700 m above ground level).

The strong plumes from the megacities of Paris and London are easily distinguished when considering the signature of anthropogenic emissions in Fig. 2b and 2c, with their amplitude exceeding 0.3 ppm at 100 km downwind of these cities, and with sharp gradients of $XCO_2$ at their edges. The relative narrowness, extended length and small intensity of the plumes shown in the Figs. 2b and 2c are explained by the magnitude of the wind speed on May the 23[rd]. The characteristics of those plumes vary considerably with respect to the wind speed and the inversion results are strongly impacted by this parameter (section 3.2.1 and Broquet et al. (2018)).

Figures 2b and 2c also show that the overlapping of plumes from urban areas in Belgium and in the Netherlands produces $XCO_2$ patterns whose amplitudes are comparable to that of the plumes from Paris and London. However, due to the urban density of those countries, the level of distinction between the individual signatures of the different cities is weak. If we exclude Paris, northwestern France has a much less dense urban fabric with scattered cities of small extents. This sparse distribution allows the relatively weak plumes from cities to be visible whereas the more diffuse $XCO_2$ signatures of the countryside emissions do not form any distinguishable patterns (Fig. 2b).

The comparison between Figs. 2b and 2c highlights the plumes from some of the large 84 point sources within the 2-km resolution subdomain (section 2.2.2). The amplitude of these plumes can locally reach that of Paris but such an increase above the background occurs on a much smaller extent: for instance, the one of the power plant close to Dunkerque (~51°, ~2.3°) in the northern French coast reaches 0.4 ppm but its width does not extend to more than 5 km (Fig. 2c). The capacity of our high resolution transport model to simulate narrow $XCO_2$ plumes from point sources or urban areas distinct from that of neighbor or surrounding sources is revealed by the example of several point sources in Belgium as well as that of the oil refinery of Grandpuits (48.59°, 2.94°) whose plume stands out of the large plume from the Paris urban area (Fig. 2c). The 2-km resolution zoom of the model grid allows distinguishing those features which would be blurred in a coarser resolution transport model.

When including the $XCO_2$ produced by the natural fluxes and the boundary conditions, identifying the features produced by the anthropogenic emissions is more difficult (Fig. 2d). The atmospheric signatures of Paris, of London and of the high-emitting power plants are hardly differentiated from patterns produced by the boundary conditions and natural fluxes even though they are still visible. The isolated plumes of low amplitudes from scattered cities with small extents and low emission budgets can hardly be seen. The boundary conditions and the natural fluxes tend indeed to produce signatures whose amplitude is often larger than, or at least comparable to that of the signal from the anthropogenic emissions, with which they interfer. This blurs this signal of the anthropogenic emissions, especially when the emissions are diffuse . Boundary conditions and natural fluxes are however much more distributed homogeneously than the anthropogenic emissions, that are localized over a small fraction of the surface. As a consequence, the boundary conditions and the natural fluxes produce smooth $XCO_2$ fields (Fig. 2d) while the anthropogenic emissions produce heterogeneous fields with finer structures and sharper gradients (Fig. 2c). Therefore, the separation between the two types of fluxes could rely on the differences in terms of spatial scales of their atmospheric signatures, or on a precise knowledge of the atmospheric transport patterns.

This first qualitative overview of the atmospheric signatures could imply that the ability to quantify the budgets of emissions for the two megacities, for most of the 84 large point sources, and for large regions in the north east should be much larger than for the individual urban areas in most of the domain or for the countryside emissions. However, this diagnostic relies on a qualitative assesment of Figure 2. In Section 3.3, we will quantitatively analyze the the inversion results as a function of the type of sources.

**3.2. Potential of the satellite images to monitor the anthropogenic emissions of a megacity: sensitivity studies**

This section assesses the performance of our inversion assimilating $XCO_2$ images to monitor the anthroprogenic emissions from the Paris area, as a function of the wind conditions, of the $XCO_2$ observation precision, resolution and swath, and of the configuration of the prior error covariance matrix **B**. Results are relative to the inversion control area that covers most of the Paris urban area (Fig. 2a). The analysis is based on 62 6-hour inversion tests with satellite images nearly centered on this

area for each day of March and May 2016. With the reference 300-km wide swath, such images cover the plumes from the Paris urban area entirely in most wind conditions (Broquet et al. 2018). In the following, the wind speed is characterized by its averaged value at 700 m above the ground level over the inversion control area corresponding to Paris and over the period corresponding to the chosen diagnostic: over 5:00-11:00 when analyzing the uncertainty in the budgets of the emissions corresponding to the full 6h-period of inversion, or over the time interval [hh,11:00] when analyzing the uncertainty in the

hourly budget of the emissions between the hours hh and hh+1.

**3.2.1. Impact of the wind speed**

A first set of inversions is conducted with the reference values for the precision, resolution and swath of the satellite observation and for the parameters of the prior uncertainty (respectively 0.6 ppm, $2 \times 2$ km$^2$, 300 km, 50% and 3 hours). These inversions are applied to 12 different days in March 2016 which present a range of average wind speeds from 2 to 14

m·s$^{-1}$ (Fig. 3). We investigate results in winter, when the amplitudes of the biogenic fluxes are low, to mitigate the influence of these fluxes, and of their variability, on the URs for the Paris emissions. Note that the time profiles modeling the variability of the anthropogenic emissions ignore day-to-day variations (except between week-end and working days) which almost removes the influence of the variability of these emissions when studying results in March only. Results are presented in terms of prior and posterior uncertainties in both the 1-hour and 6-hour budgets of emissions from the Paris urban area.

The directions of the wind are predominantly meridional so that the selection of the swath has no impact. The main analysis and conclusions in this section are similar to that of Broquet et al. (2018), so we present them briefly.

Results in Fig. 3 illustrate the fact that larger wind speeds lead to smaller uncertainty reductions for the 6h-emission budgets: on March 3 the average wind speed is 13.5 m·s$^{-1}$ and the UR with the reference values for the precision, resolution and swath is 74% while on March 10 the average wind speed ~1.8 m·s$^{-1}$ and the UR is 97% (Fig. 3a). Stronger winds decrease the UR

because, due to an increased atmospheric dilution, the amplitude of the city plume is smaller which decreases the signal-to-noise ratio for the inversion. However, when considering the UR for hourly emissions, this rule may not apply for wind

speeds lower than 6 m·s$^{-1}$. For this range of wind speeds, the posterior uncertainty in 9:00-10:00 and 10:00-11:00 emissions increases with decreasing wind speed (Fig. 3b). The inversion system shows difficulties in distinguishing the atmospheric signatures produced by consecutive hourly emissions because these signatures have a significant overlap when the wind speed is low. This explanation is confirmed by the negative correlations found between the uncertainties in consecutive hourly emissions since the magnitude of these negative correlations increases when the wind speed decreases (Fig. 3c). Important negative correlations explain also that for low wind conditions, 6h-emission budgets are better constrained even though hourly emissions can be poorly constrained. The overestimation of some hourly emissions is compensated by the underestimation of other hourly emissions.

The uncertainty reductions for 1h- and 6h-budgets are high for a large range of wind values: in all the tests here, the URs for the 6h-bugdets are above 74% and the UR for the 1h-budgets after 7:00 are above 62% (Fig. 3a). Concerning the 1h-budgets of the 5:00 to 7:00 emissions, the corresponding URs significantly decrease for wind speeds above 10 m·s$^{-1}$. In particular, the UR for the 5:00 to 6:00 emission drops below 20% above this value for the wind speed. This behavior is consistent with the fact that the signatures of emissions occurring well before the satellite overpass have been much more diffused through atmospheric transport at the observation time than that of later emissions.

### 3.2.2. Impact of the precision, resolution and swath of the satellite images

Figures 4a-d show the URs for the 6h-emission budgets of March 3 (with a strong wind), March 10 (with a low wind), May 23 (with a strong wind) and May 27 (with a low wind). This figure corresponds to a second set of inversions performed with the range of options for satellite data precisions and resolutions presented in section 2.1.2 but with the observation swath, the relative prior uncertainty in the 6h-budgets of fluxes, and the correlation length scale for the prior uncertainties in hourly fluxes fixed to the reference values (respectively 300 km, 50% and 3 hours). The sensitivity of the UR to the measurement precision and resolution increases with stronger winds. For example, on May 27 (under weak wind) and 23 (under strong wind), the UR increases by 16% and 48% respectively between inversions with 2 ppm precision data and inversions with 0.3 ppm precision data (at the resolution of 2 km × 2 km). For those two days, the UR decreases by 6% and 20% respectively between inversions with 2 km × 2 km resolution data and inversions with 4 km × 4 km resolution data (with a precision of 0.6 ppm). The comparison between results on March 3 and 10 confirms such a high sensitivity under stronger winds. It is related to the fact that the slope of the convergence of the UR towards 100% with better precision and finer resolution is smaller with low wind speeds, which generate higher UR than high wind speeds. For similar reasons, the sensitivity to the precision decreases at finer resolution, and the sensitivity to the resolution decreases with better precision (Figs. 4a-d).

The comparison between results obtained when doubling the random measurement error of the individual observations and when multiplying by four the value of their spatial resolution provides insights on the exploitation of the fine scale patterns of the XCO$_2$ image by the inversion. Indeed, both changes result in doubling the resulting error at coarse resolution, but doubling the random measurement error at fine resolution conserves the capability to exploit information at this fine resolution unlike coarsening the spatial resolution of the image. Figs. 4a-d show that scores of UR with 2 km × 2 km

resolution and 2 ppm precision data are extremely close to that with 4 km × 4 km resolution and 1 ppm precision data. URs with 2 km × 2 km resolution and 1.2, 1, 0.8 or 0.6 ppm precision data are also similar to URs with 4 km × 4 km resolution and respectively 0.6, 0.5, 0.4 or 0.3 ppm precision data. This indicates that the inversions here do not really take advantage of the information on the fine scale patterns of the plume from Paris.

A third set of inversions is conducted to study the sensitivity of the results to the width of the satellite swath while keeping

all other observation and inversion parameters to reference values. This sentivitity is modulated by the wind conditions: the speed and direction of the wind control the spread and position of the plume and thus the value of the swath which fully covers the extent over which the amplitude of the plume is significant for the inversion, i.e., the value of the swath above which the results do not change any more (Figs. 4e-h). This threshold value of the swath is lower for smaller wind speed. For wind directions across the satellite track, the URs for the 6h-emissions of the Paris area are no longer sensitive to the

increase of the swath above a value of 100 km and 400 km for wind speeds lower than 8 m·s$^{-1}$ and 9 m·s$^{-1}$ respectively. The sensitivity to the swath is null (except if considering very low values for the swath of the order of the width of the plume from Paris) for wind directions along the satellite track as we consider satellite tracks centered on Paris.

### 3.2.3. Impact of the definition of the prior uncertainties in the CO$_2$ fluxes

The prior uncertainty covariance matrix **B** has a strong influence on the scores of posterior uncertainties when its

"amplitude" is comparable to, or much larger than the one of the **H$^T$RH** matrix (see Eq. 2), i.e. once the prior uncertainties are comparable or much larger than the projection of the observation errors in the control space. The relative prior uncertainty in the 6h-emission budgets ($\sigma_{Budget}$, section 2.4.), which characterizes the diagonal of **B,** is one of the critical drivers of the relative weight given by the inversion to the prior information and to the observations.

In a fourth set of inversions, we thus analyze the sensitivity of the inversion results for 6-hour emission budgets to $\sigma_{Budget}$,

with values for this parameter ranging between 0 and 100%. This set of inversions uses the reference values of the observation parameters and for the temporal autocorrelation of the prior uncertainties. Figs. 5a-b show the corresponding results on March 3 (under strong wind) and on March 10 (under low wind), to highlight the dependence of this sensitivity to the wind speed. The curves of UR as a function of $\sigma_{Budget}$ have an inflection point for values around 50%. For low values of $\sigma_{Budget}$, the UR is sensitive to this parameter, the posterior uncertainty balancing the prior uncertainty and the projection of

the observation error. For large values, the UR converges asymptotically towards 100%, and the posterior uncertainties are dominated by the projection of the observation error (i.e. the posterior estimate of the emission essentially relies on the top-down information from the observations). The observational constraint on the inversion is larger on March 10 than on March 3 since the wind is much lower on the former. As a consequence, the qualitative threshold for $\sigma_{Budget}$ above which the URs are not much sensitive to this quantity is smaller on March 10 than on March 3: 30% and 50% respectively.

These results over Paris suggest an empirical choice of a reference value for $\sigma_{Budget} > 50\%$, in the absence of any factual knowledge about $\sigma_{Budget}$. With 50% as a reference value, we focus our analysis of the posterior uncertainties on the projection of the information from the observations and we nearly neglect the prior information, while keeping an assumption regarding

the prior uncertainties that could seem consistent or even optimistic compared to series of assessment of the errors in inventories for cities at daily scale (Wang et al., 2020). However, for other cities, for point or area sources with smaller amplitudes, the observational constraint is lower. The relative weight between the projection of the observations and the prior information is then more balanced than for Paris, and the prior uncertainty still has a significant impact on the posterior uncertainties when using $\sigma_{Budget}$=50%. In order to study the pure projection of the observation errors, results using $\sigma_{Budget}$=100% will thus be analyzed along with that using $\sigma_{Budget}$=50% in section 3.3.

The other important parameter defining the **B** matrix in this study is $\tau$ (section 2.4.). By construction, the increase of the corresponding auto-correlations in the prior uncertainties at the hourly scale in **B** does not modify the prior uncertainties in the 6h-emission budgets. However, it can help the inversion crossing the information on different hourly budgets to better constrain the overall budget of emissions. A fifth set of inversions with the reference values for the observation parameters and for $\sigma_{Budget}$ is conducted to test the sensitivity to $\tau$, with values for this parameter from 0 to 6 hours (0 h indicating that there is no temporal correlation in **B**, and 3 h being the reference value), on March 10 and 3. The analysis shows that, actually, the increase of $\tau$ hardly impacts the results for the 6h-budgets (not shown) but significantly changes the results for the hourly budgets (Figs. 5c-d). The auto-correlation brings information about the temporal distribution of the emissions, constraining how the 6h-emission budgets are distributed at the hourly scale. This impact is more significant when the $XCO_2$ signatures of the hourly emissions overlap, i.e. for hourly emissions between 5:00-7:00 when the wind speed is high and for almost all the hourly emissions when the wind speed is low. However, this better knowledge about the temporal variations from auto-correlations does not appear to improve the knowledge on the 6h-budgets.

### 3.3. Potential of satellite images to monitor anthropogenic emissions at the regional, city and local scales.

This section synthesizes the inversion results at the local (for power plants, industrial facilities) to the regional scales over most of the model 2-km resolution subdomain, using a sixth set of inversions assimilating images that cover this subdomain entirely with a 900-km swath centered on Belgium (Fig.1). This set of inversions covers all the days of March and May 2016 in order to analyze the impact of the wind speed and of the natural fluxes on the results. The prior relative uncertaintes in the 6-hour budgets of the emissions are alternatively set to $\sigma_{Budget}$=50% and 100%. These inversions use the reference parameters for the observation precision and resolution and for the temporal auto-correlation of the prior uncertainties in hourly emissions (0.6 ppm, 2 km × 2 km and 3 hours respectively). Results over most of the 2-km resolution subdomain using different observation spatial resolutions and precisions will briefly be discussed in section 4.

### 3.3.1. Overview of the inversion performance

Figure 6 gives a geographical overview of the scores of UR in the 2-km resolution sub-domain. The largest scores of UR for 6-hour budgets are obtained for the mega-cities of Paris and London with mean values over the two months considered > 80%. Mean UR can also be > 60% for several cities of Belgium and the Netherlands and for a large number of point sources

(power plants and large industrial facilities) within the dense industrial area of western Germany, although these sources are
close to each other or to other significant point and area sources.

In a general way, the scores of UR increase with the magnitude of the emissions (Fig. 7). This increase is more important when considering lower emission values due to the asymptotic convergence of the UR towards 100% for high emission values (with a point of inflection for sources of ~2 MtC·yr$^{-1}$ in the curves of Fig. 7). The increase of the UR as a function of the budgets of emissions is different if considering point or area sources. As expected, the largest URs are obtained for
narrower sources like point sources (Fig. 7a) and cities (Fig. 7b) which generate plumes with smaller extents but larger amplitudes than diffuse countryside emissions. When using $\sigma_{Budget} = 100\%$, the mean URs are larger than 50% for all point sources and cities with an emission rate larger than 2 MtC·yr$^{-1}$, while to achieve 50% UR, an emission rate of at least 4 MtC·yr$^{-1}$ is needed for regional countryside emissions (Fig. 7c). The gap is even larger when using $\sigma_{Budget} = 50\%$, with mean URs that are systematically larger than 50% for annual emission budgets of point sources and cities larger than 2 MtC·yr$^{-1}$,
but for annual emission budgets of regional countryside emissions larger than 7 MtC·yr$^{-1}$.

When aggregating the results for point sources, cities and countryside emissions at the regional scale, the relative prior uncertainty becomes significantly smaller than the values used for individual sources since we assume that there is no correlation between their uncertainties: the mean prior uncertainty for the regions is then of ~33% when assuming a 50% prior error on the 6h-budgets of point sources, cities and countryside areas which make these regions. And, the emission
threshold above which the URs for the regional budgets are larger than 50% becomes 10 MtC·yr$^{-1}$ and 7 MtC·yr$^{-1}$ when using $\sigma_{Budget} = 50\%$ and 100%, respectively (Fig. 7d). These thresholds are larger than the ones corresponding to individual point sources and cities as given above, but the overall performance of the inversion system at the regional scale is better with respect to that of the point sources and cities when analyzing the relative posterior uncertainties: for $\sigma_{Budget} = 50\%$, the mean value is 22% for the total regional budgets while it is ~40 % for the point sources and cities budgets (Fig. A1).

The results for the different types of sources are shown for four regions of Belgium in Fig. 8. This figure provides an illustration of the general results seen in Fig. 7. It shows that the URs for emissions of the largest urban areas (emitting more than 2 MtC·yr$^{-1}$) are as high as that for the overall emissions of their respective region although the budgets of emissions from these urban areas are much smaller than that of their regions. As suggested above, smaller prior uncertainties in the regional budgets lead to similar URs for cities and regions budgets even though the relative posterior uncertainties in
regional budgets are much smaller (Fig. 8b). When comparing point sources and cities which are characterized by the same prior uncertainty, the relative magnitudes of the URs are determined by the relative magnitudes of the emissions: URs are thus much higher for the largest urban areas than for point sources and cities that emit much less $CO_2$. But, the comparison between the URs and emissions of the main cities and countryside areas of the regions of Eastern and Western Vlaanderen illustrates that, even thoughthey have lower emission budgets in these regions, cities are better constrained than countryside
areas. This is in agreement with the enhanced capacity of the inversion system to monitor city emissions with respect to more diffuse emissions. This figure also qualitatively illustrates the ability of the inversion system to separate neighbor emission sources: the point source and city of Liège (left blue bar for the region of Liège in the figure) contained within the

region of Liège are characterized by significant URs even though the point source is within the city of Liege and its plume is completely overlapped by the plume from the rest of the city. We will analyze more systematically and quantitatively the capacity of the inversion to disentangle the signals produced by neighbor sources in section 3.3.3.

The URs for the 6h-emission budgets show an important variability over the 62 inversion days as illustrated in Fig. 7. When using $\sigma_{Budget} = 50\%$, the standard deviations of the day-to-day variations of the URs for the point sources, cities and countryside areas, are on average, equal to ~12%, ~8.3% and 12.2%, respectively. These values are important with respect to the temporal mean of the values of UR (26%, 16% and 27% when averaging across all the point sources, cities and countryside areas respectively). These variations are associated to variations in the wind speed at the daily scale as evidenced for the Paris case in section 3.2.1. However, when considering results for the months of March and May together, they are also driven by the time profiles of the anthropogenic emissions that are characterized by a strong decrease of emissions between March and May due to the reduction of residential heating. Moreover, the UR variability is also determined by that of the uncertainties in the natural fluxes which are also very different from March to May. The natural fluxes have large negative amplitudes in May when they are dominated by the primary production and smaller positive amplitudes in March when they are mostly restricted to the heterotrophic respiration. Using constant prior relative uncertainties in the natural fluxes (as for the anthropogenic emissions) yields large absolute uncertainties in May and low absolute uncertainties in March. Furthermore, as the primary production related to photosynthetic processes is mostly driven by the radiative forcing and then by the daily variation of the cloud cover, natural fluxes and their prior uncertainties are also characterized by a strong day-to-day variability during the month of May whereas in March, they are not because of a weak day-to-day variability of heterotrophic respiration. Cross sensitivity studies comparing the influence of the above drivers (not shown), indicate the predominant influence of the daily variability of the wind speed on the variability of UR for the anthropogenic emissions estimates, for most sources. This conclusion should however be nuanced for some regions and countryside areas where the scores of UR for the anthropogenic emission estimates is highly impacted by the inversion of the natural fluxes, and thus by the variability of these fluxes, during the month of May (see section 3.3.2. below).

### 3.3.2. Impact of the uncertainties in the biogenic fluxes

The analysis of $XCO_2$ patterns produced by the different $CO_2$ fluxes (section 3.1) suggests that the large signatures of the biogenic fluxes in May could impact the monitoring of the anthropogenic emissions. In order to weigh the impact of the uncertainties in biogenic fluxes, we conduct experiments where these uncertainties are ignored. In these experiments, the mean URs for the budgets of the regional and countryside anthropogenic emissions in May are equal to ~31% and ~41% respectively (using $\sigma_{Budget}=50\%$). When accounting for uncertainties in biogenic fluxes, these mean URs decrease down to ~21% and ~31% respectively (Figs. 9c-d). This reveals some difficulty of the inversion system to separate countryside emissions from biogenic fluxes which is also illustrated by the important anti-correlations (-40% on average) between the corresponding posterior uncertainties. During May, the smaller amplitudes and rather diffuse nature of countryside emissions with respect to natural fluxes (not shown), and the overlapping of their atmospheric signatures (Fig. 2) explain why the

inversion system only has a limited ability to distinguish the countryside emissions. Oppositely, during March, the smaller amplitude of natural fluxes compared to countryside emissions (not shown) explains why the inversion can better filter the signature of countryside emissions from that of natural fluxes.

Contrarily to the URs for countryside emissions and regional budgets, the URs for the point sources and cities are hardly
impacted by the uncertainties in biogenic fluxes during the month of May, even when the emission budgets of these sources are smaller than 1 MtC·yr$^{-1}$ (Figs. 9a and b), and even though these budgets are quantitatively lower than the absolute value of the regional budgets of biogenic fluxes. Consistently, the posterior uncertainty in the inverted 6-hour budgets of the emissions of cities or point sources is weakly correlated with that in the 6-hour budgets of biogenic fluxes in their respective region (-2% and -5% on average for the cities and point sources respectively). Therefore, the visual inspection of Fig. 2 may
wrongly suggest that the plumes of the smallest point sources and cities controlled individually can hardly be separated from the signature of biogenic fluxes. The differences in terms of spatial scales of the atmospheric signatures appear to be the main driver of the skill of the inversion to separate anthropogenic sources from biogenic fluxes.

### 3.3.3. Separation of the different anthropogenic emission sources

In order to estimate the ability of the inversion system to monitor anthropogenic sources whose atmospheric signals overlap
separately, we focus on pairs of sources contained within the same region and we assess whether the sum of the variances (*Var*) associated to the inverted emissions for each source is comparable with the absolute value of their covariance (*Cov*), which is nearly systematically negative as a result of the uncertainty in the spatial attribution of emitted $CO_2$ to individual sources in the inversion. This criterion means, if the covariance is negative, that the variance associated to the ensemble of the two sources is much smaller than the sum of the variances associated to each source, since that the variance of the sum of
two random variables $X_a$ and $X_b$ is given by $Var(X_a + X_b) = Var(X_a) + Var(X_b) + 2 * Cov(X_a + X_b)$. In terms of inversion, this case describes the situation when a pair of sources is much better constrained than each of its individual sources ($Var(X_a + X_b) \ll Var(X_a) + Var(X_b)$), *i.e* when the inversion system does not entirely manage to disentangle overlapping signals and to constrain independently each source.

Figure 10 indicates that in general, the system has a good ability to constrain independently the different sources within a
given region: point sources, cities, countryside areas, natural fluxes. In this figure, indeed, the number of pairs of sources that are characterized by an important negative covariance term ($2 * Cov(X_a + X_b)$) compared to the sum of the individual variances ($Var(X_a) + Var(X_b)$) is much lower than the number of pairs of sources that are characterized by a relatively small covariance term. Only 20 pairs of sources out of a total of 890 pairs show a covariance term that is larger than 25% of the sum of the variances (squares below the blue line in Fig. 10a). Most pairs for which the inversion system may be unable
to distinguish individual sources in a completely independent way consist in a point source and its surrounding urban or countryside area. Only one case consists in two point sources that are located close to each other near Karlsruhe in Germany and, no cases consist in pairs of urban/urban or urban/countryside areas. The 20 cases of less separable pairs of sources identified above could be associated to situations for which the inversion system could have difficulties to independently

monitor two sources but this conclusion should be nuanced when all terms (sum of the variances and covariances) are small: for example, four pairs of sources characterized by important negative correlations show an UR for each individual sources which is larger than 50% (squares in the top-right quadrant of the Fig. 10b). For these cases, the problem of distinction between the two sources applies to a moderate residual posterior uncertainty, and the inversion still gets a relatively precise estimate of the emissions.

Moreover, a low covariance between sources can often be explained by the lack of constraint on their total budget or on one of the two sources, rather than by a good separation between the two sources. Examples of pairs of sources with an important UR for their total emissions and a small covariance while only one individual source in each pair is well constrained can be seen in the top-left and bottom-right of Fig. 10b (where the UR for the total is larger than 75% while one of the sources shows an UR lower than 20%). Pairs of sources characterized by relatively low values of covariance but small values of UR for both the total and the two sources are identified in the top-right quadrant of Fig. 10a and the bottom-left quadrant of Fig. 10b. These pairs are characterized by low individual sources and total emissions whose amplitudes are below the thresholds required by the inversion system to produce reliable estimates (see section 3.3.1.).

## 4. Discussion

Our series of tests and analysis demonstrate the potential of our inversion system to explore various aspects of the estimation of anthropogenic $CO_2$ emissions based on satellite $XCO_2$ spectro-imagery.

*A performance simulation tool*

The analytical formulation of the inversion makes it easy to test different scenarios of observation systems that monitor anthropogenic emissions but, in this study, the OSSEs with the inversion framework have been performed in order to investigate the potential of the satellite imagery of $XCO_2$ from helio-synchronous orbits only, depending on instrumental configurations or other factors such as the wind or the amplitude of the emissions. Such a spaceborne imagery similar to CO2M may become the critical component of operational atmospheric inversion systems for the monitoring of $CO_2$ anthropogenic emissions. These operational inversion systems will likely have to jointly assimilate data from many of the existing satellite missions, from ground-based networks and from the planned spectro-imagers. Moreover, in addition to $CO_2$ concentrations, co-emitted pollutants and radiocarbon should be assimilated as well. Ultimately, OSSEs would have to integrate all these components of the observation system (Ciais et al., 2015; Pinty et al., 2017). However, we need OSSEs considering the spaceborne imagery of $XCO_2$ as a stand-alone system to determine the instrumental parameters ensuring that this imagery can bring emission estimates with sufficient coverage and accuracy so that it can serve as a backbone for the operational monitoring of emissions. Furthermore, in a context where there is a lack of ground based and spaceborne

networks that are suitable for the monitoring of $CO_2$ anthropogenic emissions, these OSSEs can help better understand the needs in terms of complementary observation components.

The analytical inversion system built for this study allows testing an important set of observational parameters and situations. Once the atmospheric transport functions associated to the different sources have been computed, the derivation of the posterior uncertainties is fast and results can be delivered for a wide and detailed range of specifications on the spatial resolution, precision (focusing here on the errors in $XCO_2$ from the instrumental noise) and swath of the satellite $XCO_2$ images. Extending the study of Broquet et al. (2018), the sensitivities of the inversion results to these three parameters have been tested on the example of the inversion of the emissions of Paris (section 3.2.2.). We varied these parameters without accounting for current limitations in space technologies (or from cost issues) which impose potential trade-offs between their configurations. The impact of the wind speed on the determination of the 6-hour and hourly budgets of the emissions is analyzed by assimilating images over different days of March and May. Finally, the flexibility of the inversion system also allowed testing the impact of the way prior uncertainties in the fluxes are prescribed.

The inversion framework produces curves of sensitivity that give several qualitative and quantitative insights into the optimal configurations of the satellite imagery: the study of the UR for the Paris emissions indicates (i) that a 4 km × 4 km spatial resolution is sufficient, provided that the precision at this spatial resolution is very high, and (ii) that the inversion hardly makes use of patterns at finer spatial resolution. However, as discussed in section 3.2, that might come from the relatively large extent of the plume from Paris, and from the distances between this plume and other major plumes. We have thus extended the analysis of the sensitivity to the observation spatial resolution and precision to all the days of March and May and to the ensemble of controlled regional, city and local sources (as for the analysis in section 3.3) testing a reduced set of resolutions and precisions. The results are displayed in Fig. 11. This experiment confirms that results for point sources and narrow cities are more sensitive to the availability of the information at the reference resolution of 2 km than that for Paris. The UR for point sources and cities emitting less than 2 MtC·yr$^{-1}$ is larger with a spatial resolution of 2 km × 2 km and a precision of 1.2 ppm than with a spatial resolution of 4 km × 4 km and a precision of 0.6 ppm, (i.e. with the same precision at 4 km resolution but without information about the patterns at scales finer than 4 km). However, the differences are not really significant, and Fig. 11 tends to confirm that a spatial resolution of 4 km × 4 km would be fine if achieving a very good precision: typically, 0.3 ppm if willing to get the results from the reference configuration of the observation with 2 km resolution and 0.6 ppm precision. With a 2 km × 2 km resolution, the mean URs for all type of emissions would increase on average by 10.1 ±1.5% if the precision increases by a twofold factor (from σ = 1.2 ppm to 0.6 ppm and from 0.6 ppm to 0.3 ppm). While this represent a dramatic increase of the UR when changing the precision from 1.2 ppm to 0.6 ppm, the relative impact is smaller from 0.6 to 0.3 ppm and the reference value of 0.6 ppm precision at 2 km resolution appear to be a balanced option. Finally, the results for the Paris case point out that, at the reference precision of 0.6 ppm, the performance of the inversion system would be nearly equivalent at the resolutions of 2 km × 2 km and 2 km × 3 km (Fig. 4). The spatial resolution could thus be relaxed to 2 km × 3 km compared to the reference resolution without degrading much the precision of the inversions.

If we consider the monitoring of the Paris emissions, the impact of the swath on the results is limited for swaths larger than 400 km (section 3.2.2 and Fig. 4). As the megacity of Paris produces plumes whose intensity and extent are amongst the highest with respect to anthropogenic emissions sources in Western Europe, this supports the use of a swath close to 400 km to ensure that the full extent of plumes from targeted sources near the center of the satellite field of view is caught. However, a narrower swath would be sufficient for the large majority of cities which emit less than Paris, and thus have shorter plumes. This would impact the monitoring of the large megacities emitting more than 10 MtC/yr like Paris, which represent around 9% of the emissions from cities and power plants over the globe (Wang et al., 2020), but this impact is limited for Paris in most meteorological conditions as long as the swath is larger than 250 km (90% of the cases for the 62 days of inversion, not shown). A general conclusion from these results is that the swath is a less critical parameter than the pixel precision and resolution.

Overall, these results support the reference configurations for the pixel resolution and precision and for the swath width that broadly correspond to the current ones for the CO2M mission, even though they suggest a relaxation of the spatial resolution to 2 km x 3 km and an extension of the swath up to 400 km. However, since this study does not account for technical and cost trade-offs between these parameters, it cannot provide a full assessment for their combination. Furthermore, by flying a satellite any day over the areas of interest in the sensitivity tests, we restrained the role of the swath to covering a more or less important portion of the plumes, with a lack of sensitivity when the swath exceeds the plume length or when the wind blows along the satellite track. In practice, a critical role of larger swaths (like that of using more satellites in a constellation) is to increase the number of situations for which the plume from a given city can be seen, which is ignored in this study. Finally, the spatial resolution could also play a role when accounting for cloud cover since it could help to increase the spatial coverage. A more realistic simulation of the observation sampling by specific missions or constellations, with one or several satellites following real tracks and including the impact of cloud coverage and large aerosol loads, and some realistic technical constraints between the parameters, is needed to fully assess the right balance between higher precision, finer spatial resolution and larger swaths.

*Ability to monitor anthropogenic emissions at the regional, city and local scales*

At first, a qualitative overview of the atmospheric signatures could imply that the ability of the inversion system to quantify the budgets of emissions from most of the 84 large point sources and from the two megacities of Paris and London should be much larger than for smaller cities, and that countryside emissions should hardly be constrained by the inversion (section 3.1). However, the quantitative analysis of the results from the OSSEs described in section 3.3. shows that the capacity of detection of the inversion system is equivalent for point sources and cities: uncertainties in both type of sources emitting more than 2 MtC·yr$^{-1}$ are reduced on average by more than 50% when the resolution and precision of the satellite data are equal to 2 km $\times$ 2 km and to 0.6 ppm respectively, and when the prior uncertainties in the fluxes have been set to 50% of the 6 h-budgets (section 3.3.1). With the same parameters, the threshold on the level of emissions to get such an UR is much

higher (~10 MtC·yr$^{-1}$) for the countryside areas and for the whole regions. However, for these types of sources, the overall level of performance of the inversion system is still comparable to that for point sources and cities: for the 67 regions considered in this study, the mean URs for the total emissions and for the countryside emissions are of 37% and 27.4% respectively (section 3.3.1.). The lower performance of the inversion system to monitor the countryside and regional anthropogenic emissions could be partly related to the impact of the biogenic fluxes on the determination of these types of emission sources contrary to the point sources and cities (section 3.3.2). Beside these sensitivities to the source amplitudes and to the uncertainties in biogenic fluxes, the variations in URs are also driven by the wind (sections 3.2.1 and 3.3.1) and by the potential loss of part of the atmospheric signatures of sources at the edges of the satellite swath (section 3.2.2).

We have also investigated the capacity of the inversion system to deal with the overlapping of $XCO_2$ plumes produced by nearby sources (section 3.3.3). As satellite images give snapshots of the $XCO_2$ distribution at a given time (11:00) and as atmospheric transport mixes the contribution of the different sources over the 6 hours before the satellite overpass, it is important to assess the ability of the inversion system to disentangle the information coming from different sources at different times. With the hypothesis adopted in this study, most point sources and cities can be monitored independently by the inversion system. This ability could be partly related to the high-resolution modeling that allows describing and catching the fine scale patterns of the $XCO_2$ signatures of the point sources even if the inversion is weakly sensitive to patterns at scales < 4 km (Fig. 11).

Beyond the capacity to monitor separately the emissions of point sources, urban and countryside areas, the additional benefit of controlling separately these emissions within a given region is the mitigation of the aggregation errors when inverting the total budget of a region. In analytical inversions, these errors arise when a control of the emissions at a too coarse resolution limits too much the ability to fit the actual spatial distribution of the emissions and of the concentrations. These errors are evidenced for our study by inverting regional budgets without considering any internal separation into cities, point sources or countryside areas in the control vector: with such a configuration, the results are significantly and wrongly more optimistic than the ones obtained with the inversions that use this subdivision (Fig. A2).

*An optimistic framework*

This study aims at assessing the projection of the sampling and observation random noise into the emission estimates. It combines optimistic assumptions that prevent from assuming that the level of posterior uncertainties achieved in the OSSEs here should correspond to that of the inversions with real data. The ability of the inversion system to separately monitor point sources and cities (section 3.3.3) and to quantify regional budgets of diffuse emissions is partly due to the underlying assumption made in this study that the atmospheric transport is perfectly known. We ignored the transport modeling errors and implicitly supposed that the position and extent of the $XCO_2$ plumes from cities or point sources are perfectly simulated. Accounting for the transport modeling errors by assuming that they can be described as a random noise uncorrelated in space and time, as is usually done in atmospheric inversions, would not fundamentally change the results of this study. The impact

of such an error would be equivalent to decreasing the precision of the observations. However, as a consequence of errors in the wind speed and direction, or in the $CO_2$ transport model itself, the simulation of narrow and localized plumes can poorly match the actual ones and strongly affect the inversion of the emissions from point sources and cities. The actual signature of diffuse emissions could also be different in terms of shape and extent from the actual one. An inversion procedure that would simultaneously control both transport parameters and $CO_2$ emissions within a coupled meteorological-$CO_2$ transport model (Kang et al., 2011) may partially address these issues although it would greatly increase the complexity of the inversion, in particular by introducing potentially large non-linearities in the observation operator. New methods based on imagery processing (Corpetti et al., 2009), plume detection (Kuhlmann et al., 2019), Gaussian plume modeling (Nassar et al., 2017) or direct computation of fluxes through detected plumes may also help overcome realistic transport uncertainties to invert the emissions corresponding to point sources and cities (Varon et al., 2018). However, these methods may hardly deal with diffuse emissions whose signatures have low amplitudes, and should be difficult to detect in $XCO_2$ images. The recovering of countryside emissions and of regional budgets of the emissions may thus be optimistic in this study.

Moreover, the configuration of our inversion system uses many of the traditional assumptions of atmospheric inversions among which the rather simple characterization of the sources of uncertainties with Gaussian distributions, which may underestimate their impact. The perfect knowledge of the statistics of the different source of uncertainties that are accounted for is another traditional optimistic assumption used in atmospheric inversions with synthetic data and in the experiments here. Furthermore, some sources of observation errors, such as the systematic errors which bear spatial correlations, are purposely ignored in the present study. Such errors exacerbate the problem of the identification of the signatures of point source, city to diffuse emissions in addition to have a larger error budget than random noise at the spatial scales of such signatures. There is a clear need to assess the impact of such correlated errors. Lastly, the extent of the observation sampling is made rather optimistic by ignoring cloud cover and the loss of data due to large aerosol loads. In a general way, the results of this study could be seen as optimistic and as an upper limit of the skill of the inversions using satellite images only but, also as good indicators of the sensitivity of the uncertainty reduction to various parameters and drivers.

Broquet et al. (2018) provided insights into the impact of cloud cover and systematic errors in the $XCO_2$ images. They used realistic simulations of satellite samplings and errors made for the CarbonSat mission by (Buchwitz et al. 2013a). Results have indicated that, when accounting for cloud cover, the satellite data could efficiently constrain emissions from Paris only ~20 days per year, and that the impact of the systematic errors anticipated for that mission is such that the system would hardly be able to reduce errors in the emission estimates if these errors are not filtered or controlled for. However, efforts have been made to limit the amplitude of systematic errors in the concept of the new CO2M mission.

Our new inversion framework allows accounting for a realistic simulation of the observation sampling and errors. Nevertheless, generating simulations of the systematic errors from the retrieval of $XCO_2$ data that are suitable for the purpose of our study would have been difficult. Systematic errors are not described in the uncertainties computed by existing retrieval schemes. Furthermore, they depend on specific measurement configurations and on the evolving skill of radiative transfer inverse models and of empirical bias-correction systems, so that their characterization based on diagnostics with

existing missions may hardly apply to future ones. Simulating realistic patterns of cloud cover consistent with the meteorology for the different test cases investigated would have also been challenging. Finally, this study focuses on other parameters to allow exploring the sensitivity of the inversions to these parameters in depth.

In order to raise insights into the impact of errors with spatial patterns such as model and systematic errors, we have conducted experiments where spatial correlations are included in the observation error (in the **R** matrix). We have tested isotropic and homogeneous spatial correlations exponentially decaying with distance, using various correlation lengths. The experiments and results are described in Appendix A since they are out of the scope of this study. The results indicate that including correlations in the observation errors tends to increase the budget of observation errors and thus to increase the posterior uncertainties in the flux estimates as long as the correlation length scale does not exceed that of the signature of the fluxes in the $XCO_2$ images. However, including correlations in the observation errors also tends to increase the ability to distinguish between the patterns of the observation error and of the signatures of the fluxes, and thus to decrease the posterior uncertainties, so that for large spatial correlation lengths, increasing the correlation length leads to a decrease of the posterior uncertainties. In our tests, the worst situation for the monitoring of the emissions in the study area corresponds to ~10 km correlation length scales. These results should be interpreted cautiously since the spatial patterns of the model and systematic errors are more complex than this traditional but simple modeling of spatial correlations and since in these tests, the inversion system if perfectly informed about the statistics of the observation error. Future studies will integrate more realistic simulations of observation sampling and errors from different concepts of spaceborne imagery, based on radiative transfer inverse modeling applied to realistic fields of surface and atmospheric conditions and instrumental specifications.

A last significant simplification of the general problem of the inversion of the anthropogenic emissions based on $XCO_2$ data has been stressed by Ciais et al. (2020). Anthropogenic emissions of $CO_2$ bear a major share of emissions from biofuel combustion which can hardly be separated spatially from the fossil fuel combustion component. Furthermore, the emissions of $CO_2$ by human respiration represent a significant portion of the total $CO_2$ emitted from cities. The $XCO_2$ data and the atmospheric inversion approaches can hardly be used to distinguish between these different components if it cannot rely on complementary data. This factor was ignored here, as well as in most of the studies dedicated to the inversion of anthropogenic $CO_2$ emissions at city to regional scales.

*Exploiting further capabilities of the inversion framework: potential of complementary observation systems and results at larger temporal scales*

Our analysis is restricted to a window from 5:00 to 11:00 corresponding to the period which might be constrained by the satellite observation from a heliosynchronous satellite with 11:00 local overpass time. This provides little information about the capacity of the inversion system to monitor daily to monthly budgets. To address this need and improve the results of the inversions, the modularity of our data assimilation framework could integrate multiple streams of data in order to increase the temporal and spatial coverage of the information (Moore et al., 2018; O'Brien et al., 2016).

845 In future studies, our inversion system could also integrate more realistic hypotheses concerning the description of the prior uncertainties in the emissions. Uncertainties in inventories are indeed difficult to characterize and this study assumed, rather arbitrarily, that the relative uncertainty in the 6-hours budgets was of 50% or 100% (section 2.4). We also assumed that the prior uncertainties had low temporal correlations, i.e. correlation over timescales of less than 6h, and we neglected the spatial correlations between the budgets for the different point sources, cities and countryside areas. Actual anthropogenic emissions

850 and inventories have complex cycles at daily and weekly scales together with a large temporal and spatial variability. There is still a critical lack of knowledge and of characterization of the correlations in the uncertainties in the inventories (Wang et al. 2018, Wang et al., 2020). However, some extensive analyses are now conducted to fill this gap (Super et al., 2020). We can expect some stronger spatial and temporal connections than assumed in this study, which would increase the transfer of information from the atmospheric observations, *i.e.* both the uncertainty reduction for the sources and time windows covered

855 by the observations, and the uncertainty reduction for other sources, time windows, and for large spatial and temporal scales.

**Conclusion**

We have presented a new and comprehensive high-resolution atmospheric inversion system to assess the potential of the satellite imagery of $XCO_2$ for the monitoring of anthropogenic emissions from local to national scales. This system has been designed to deal with a wide range of different sources in terms of emission amplitude, distribution and spatial scale and to

860 account for the overlapping between the signatures from different anthropogenic sources and natural sources and sinks. To cover the local and regional scales while mitigating computational costs, this inversion system is based on an atmospheric transport model with a zoomed grid whose horizontal resolution ranges from 2 km in the most resolved domain to 50 km at its edges. The area of highest horizontal resolution (2 km) encompasses regions where the hourly emission budgets of a large ensemble of cities and point sources are inverted separately, *i.e.* that of northern France, Belgium, the Netherlands, western

865 Germany, Luxemburg and London. Urban areas are delimited using a clustering algorithm and large industrial plants are selected based on an emission threshold. In total, the system controls the hourly budgets of 303 different anthropogenic sources (plants, cities or regions), the hourly budgets of biogenic fluxes for 67 regions and a scaling factor applied to the initial, lateral and top boundary conditions for each inversion window.

The study investigates the potential of imagery of $XCO_2$ from satellites on helio-synchronous orbits, similar to CO2M but

870 varying the spatial distribution, precision and swath of the imagery. The aim is to analyze the relative sensitivities to these different parameters and thus to raise insights for the optimal configurations of new satellite imagery concepts. In the synthetic framework of our experiments, we simulate satellite overpasses over the area of interest at 11:00 (local time), and we conduct inversions with windows covering the 6 hours before these overpasses (5:00-11:00), that have been identified as the periods which might be directly constrained by the satellite observations. Focusing on the instrumental noise, the analysis

875 of the sensitivity to the errors on $XCO_2$ data ignores systematic errors from the retrieval process. All images are moreover assumed to be taken in clear-sky conditions with the swath centered on the targeted sources. Another critical point is that the

transport modeling errors are ignored. Finally, i) the optimal spatial coverage of the targeted plumes which are seen entirely except during the tests of sensitivity to the swath width, ii) the lack of systematic and transport modeling errors, and iii) the assumption that the inversion system is perfectly informed about the statistics of the errors in the prior estimates of the emissions, biogenic fluxes and transport model boundary conditions, raise optimistic scores of URs. However, these scores are mainly analyzed in terms of upper bound for what can be achieved by the assimilation of the satellite images, or in a relative way when diagnosing sensitivities to various parameters.The results of the sensitivity studies support specifications for the pixel resolution and precision, and the swath width that are closed to that considered for CO2M: combinations of 2 km x 2km to 2 km x 3km pixel spatial resolution, 0.6 ppm precision and 250 to 400 km swath. More constraining options do not appear to raise improvements in the precision of the emission estimates that are significant enough. The inversions illustrate a higher skill to monitor individual point sources and dense cities. However, despite their optimistic configuration, they also indicate that scores of URs of more than 50% for the 6-hour emissions before the satellite overpasses can be achieved for cities and point sources emitting more than 2 MtC·yr$^{-1}$ only (with a configuration where the resolution and precision of the satellite data are equal to 2 km $\times$ 2 km and to 0.6 ppm respectively, and where the prior uncertainties in the 6-hour emissions have been set to 50%).

Exploiting the modularity of our inversion system and coupling it to radiative transfer inverse models, future studies should be conducted to assess the performance of more specific concepts of spaceborne imagery under more realistic scenarios for the sampling, for the XCO$_2$ retrieval errors and for the could cover. They should also analyze the complementarity of the imagery from helio-synchronous orbits with ground-based measurements or with different types of spaceborne observation sensors.

**Appendix A: testing the impact of spatial correlations in the observation error**

When addressing the impact of errors from the atmospheric modeling and observation, our study focuses on the random error from the satellite instrumental noise which is assumed not to bear spatial correlations (following, e.g., Buchwitz et al., 2013a). It does not study the impact of two major sources of uncertainties in the inversions: the transport modeling errors and the systematic errors in the XCO$_2$ retrievals from the inverse radiative transfer modeling (Hobbs et al., 2017). These errors raise spatial correlations in the overall observation errors (Worden et al., 2017; Broquet et al., 2018).

Beside the need to limit the scope of the study to a reasonable extent, one reason for not studying the impact of these errors here is that they depend on the specific modeling and measurement configurations and on the evolving skill of transport and radiative transfer models and of empirical bias-correction systems (see, e.g., the dramatic improvement of the agreement between NASA's Orbiting Carbon Observatory-2 retrievals and reference ground-based retrievals between version 7 and version 8 of NASA's algorithm in Fig. 18 of O'Dell et al., 2018). Therefore, conclusions from existing missions may hardly apply to future ones. Another reason is that there are characterized empirically and that we still lack of robust and theoretical

ways to describe them. However, in order to feed the discussion and perspectives regarding such errors, we provide here a first exploration of the impact of spatial correlations in the observation errors.

A traditional way to model spatial correlations between observation errors is to assume that they decrease with the distance, and that they are isotropic and homogeneous in terms of spatial scale (Chevallier, 2007). Therefore, we model such correlations using exponentially decaying functions (exp(-d/D)) of the distances d between two observation pixels, the parameter D defining the correlation length scale. Their inclusion in **R** (sections 2.3.1) is tested by applying such correlations to the total observation errors, i.e., not defining the total observation errors has a combination of spatially correlated and

uncorrelated errors, but as a single component with spatial correlations. This choice eases the analysis of the impact of spatial correlations here, which aims at raising first general insights on the behavior of the inversion when including them rather than at providing a quantification of such an impact.

In a new set of tests with our inverse modeling framework, we use the set-up which covers the sources over most of the 2-km resolution model subdomain based on 900 km wide XCO$_2$ images (section 3.3). We set **R** with a 1-sigma uncertainty of 0.3

920  ppm for all XCO$_2$ data, and with D=10, 50 or 100 km. In order to lighten the computations associated with the inversion of a matrix **R** that is no longer diagonal (Eq. 2), we perform the inversions by considering observations with 4 km resolution pixels rather than 2 km resolution ones. The choice of a data precision of 0.3 ppm allows thus to get results for a configuration that is close to the reference one (with data resolution and precision of 2 x 2 km$^2$ and 0.6 ppm respectively; see section 3.2.2 and 4 regarding the similarity of the results from the inversions done with a given instrumental precision and

resolution and the inversions done with an instrumental precision twice smaller and a resolution four times larger).

When introducing correlations with small spatial scales (D=10km), the uncertainty reductions for all type of emission sources are lower than for the inversions performed without correlations within the **R** matrix (Fig. A3). When using D=50km, this is still true for most of the cities and for countryside and regional emission areas. However, the uncertainty reductions get larger than when ignoring spatial correlations for all point sources (except 2) and the cities that emit less than

~25MtC/year. Finally, when using D=100km, the uncertainty reductions get larger than when ignoring spatial correlations for all point sources, all cities emitting less than 40MtC/year, and a significant number of countryside and regional emission areas. (i) This behavior, (ii) the analysis of images of the plume from Paris with the different types of observation errors tested here (Fig. A4), (iii) the fact the cities with the largest emission rate are generally also those with the largest spatial extent, and (iv) the fact the sources with larger emission rates have an atmospheric signature which can be distinguished on

larger spatial extent, lead us to the following interpretation of the impact of the spatial correlations in **R**:

- when introducing correlations with small spatial scales in **R**, the posterior uncertainties in flux estimates increase since these correlations yield larger budget of observation errors at the scale of the signatures of the targeted fluxes. This impact tends to saturate when the spatial correlation scale in **R** reaches and gets larger than the scale of the signatures of the targeted fluxes.

- conversely, the increase of the correlation scales helps the inversion separate the observation error patterns from these signatures.

- These two opposed effects lead to a worst case in terms of posterior uncertainties in the emission estimates that correspond to correlation scales in **R** that are function of the scale of the flux atmospheric signatures. In particular, it varies depending on whether we analyze results for point sources, cities or widespread emissions across regions.

These insights into the impact of spatial correlations in **R** call for further investigations. In particular we should test more complex patterns that could better correspond to actual model and systematic retrieval errors (e.g. following surface and atmospheric structures). We should also challenge the potential of the inversion to separate such structured observation errors from flux atmospheric signatures in conditions where this inversion is not perfectly informed about the error spatial correlations.

**Acknowledgements**

This work was supported by the Chaire Industrielle Trace ANR-17-CHIN-0004-01 cofunded by the ANR French national research agency, THALES ALENIA SPACE, SUEZ-Environnement, and TOTAL-Raffinage Chimie. The authors are grateful for TAS partners, in particular Sandrine Mathieu, LMD partners, in particular Vincent Cassé, Cyril Crevoisier and Olivier Chomette, and colleagues from LSCE: Yilong Wang, Elise Potier and all the Chaire TRACE team for fruitful

discussions. This work was performed using HPC resources from GENCI-TGCC (grant no. 2019-A0070102201).

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

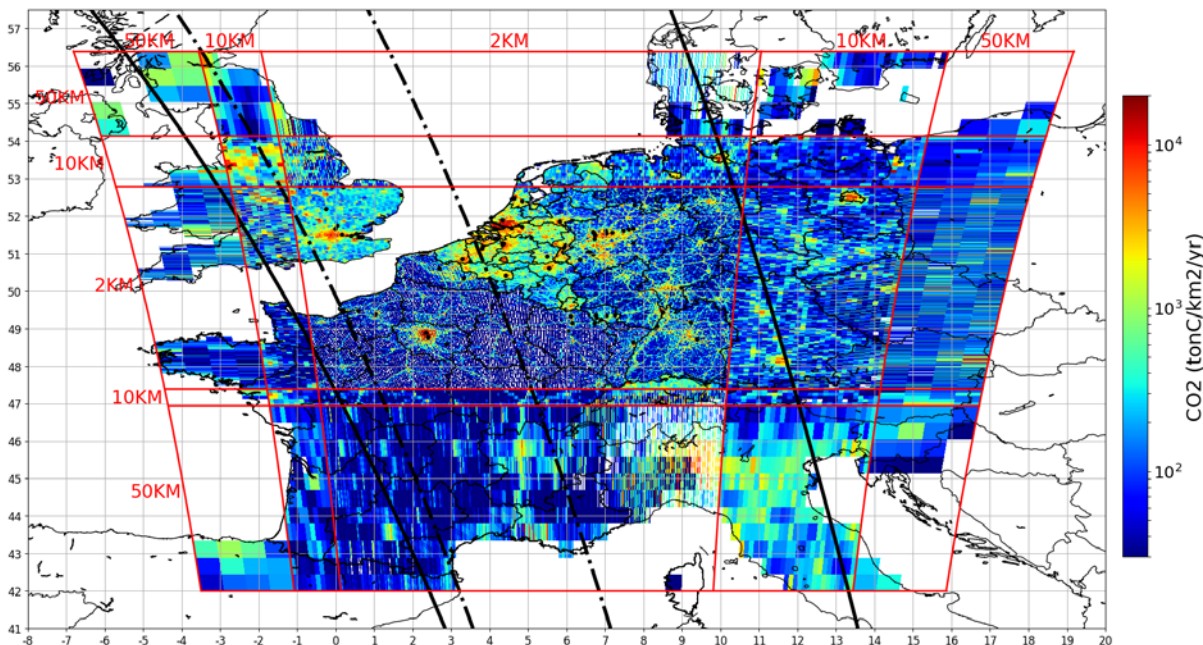

**Figure 1: Maps of the IER annual emissions interpolated over the domain of the CHIMERE model. Point sources are indicated by black dots and administrative regions by thin black lines. The grid of the model is defined by sub-domains with several resolutions (r×s km² where r and s=2, 10 or 50 km) and whose boundaries are represented by the red lines. The thick black lines depict the edges of the satellite tracks corresponding to the synthetic data used in this study (300 km swath: dash-dotted line, 900 km swath: solid line).**

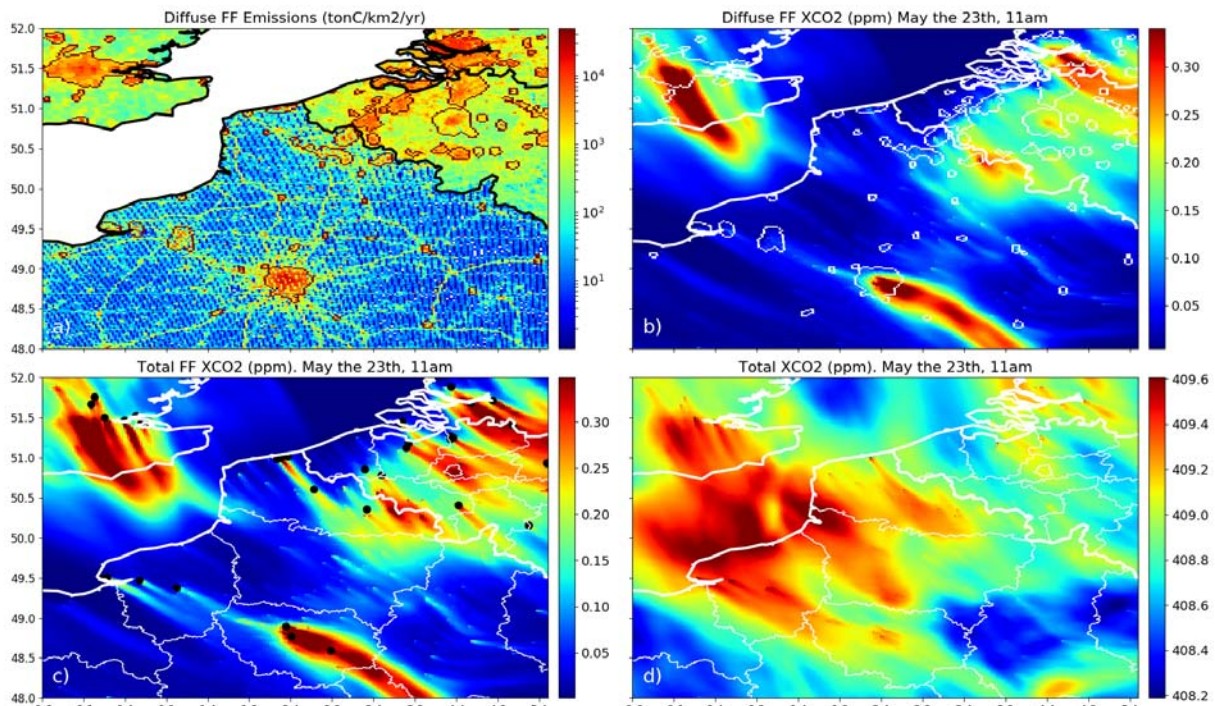

**Figure 2: IER emission maps interpolated over northern France, western Belgium and the London area (a). We have represented the anthropogenic emission sources without point sources ("area emissions"). Red curves depict the boundaries of the city clusters defined by a pattern recognition algorithm (Section 2.2.3.). Panels (b) and (c) show the simulations of XCO$_2$ (ppm) on May the 23$^{rd}$ at 11am that are produced respectively by the area and total anthropogenic emissions between 5:00 and 11:00. Point sources are indicated by black dots in panel (c). In panel (d), simulations of XCO$_2$ (ppm) on May the 23$^{rd}$ at 11am that are produced altogether by the anthropogenic, natural and boundary fluxes between 5:00 and 11:00. For the sake of clarity, these figures do not show the whole 2 km-resolution sub-domain of CHIMERE, but illustrate the patterns seen over this subdomain well.**

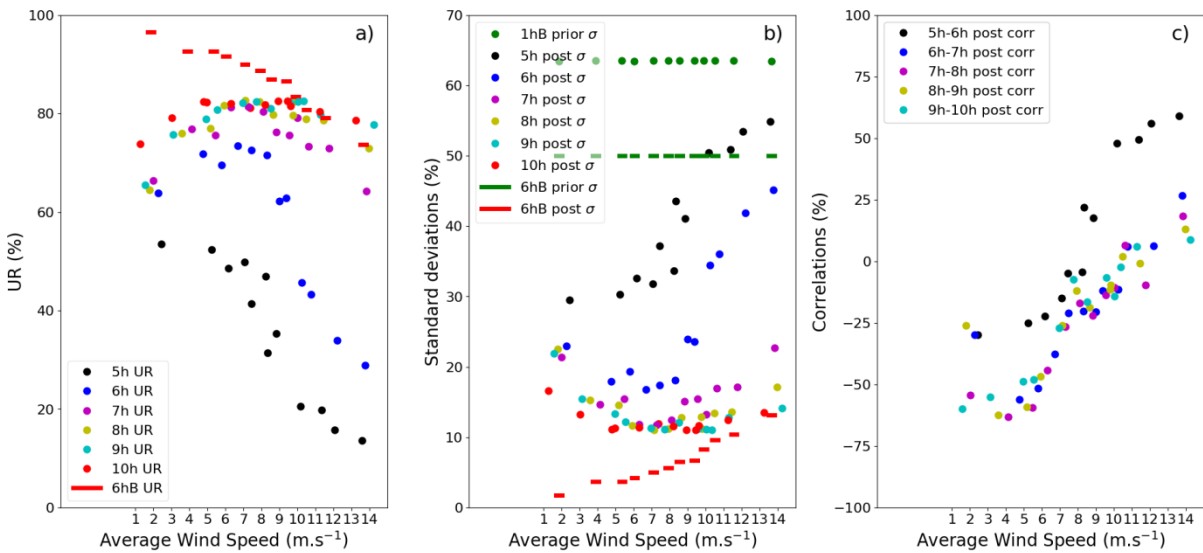

**1095**

**Figure 3: Uncertainty Reductions (UR) from a 50% prior uncertainty on the 6h-budgets of the Paris emissions for 12 days characterized by different average wind speeds over Paris (a). UR for hourly and 6h-budgets of the Paris emissions are shown by color dots and red segments respectively. In panel (b) are shown prior *vs.* posterior uncertainties on 1h-emissions (color dots) and 6h-emissions of Paris (green and red segments). In panels (a,b), the colors of the dots represent the hour of the corresponding 1 h-budget; the green dots are for the prior uncertainties on the 1 h-emissions (1hB prior σ) which are derived from 50% prior errors on the 6h-budgets (6hB prior σ) and by considering temporal prior correlations of 3 hours. Panel (c) shows correlations between posterior uncertainties in 2 consecutive 1 h-emissions (color dots). Results are computed with a retrieval resolution of 2 km×2 km, a precision of 0.6 ppm and a swath of 300 km.**

**1100**

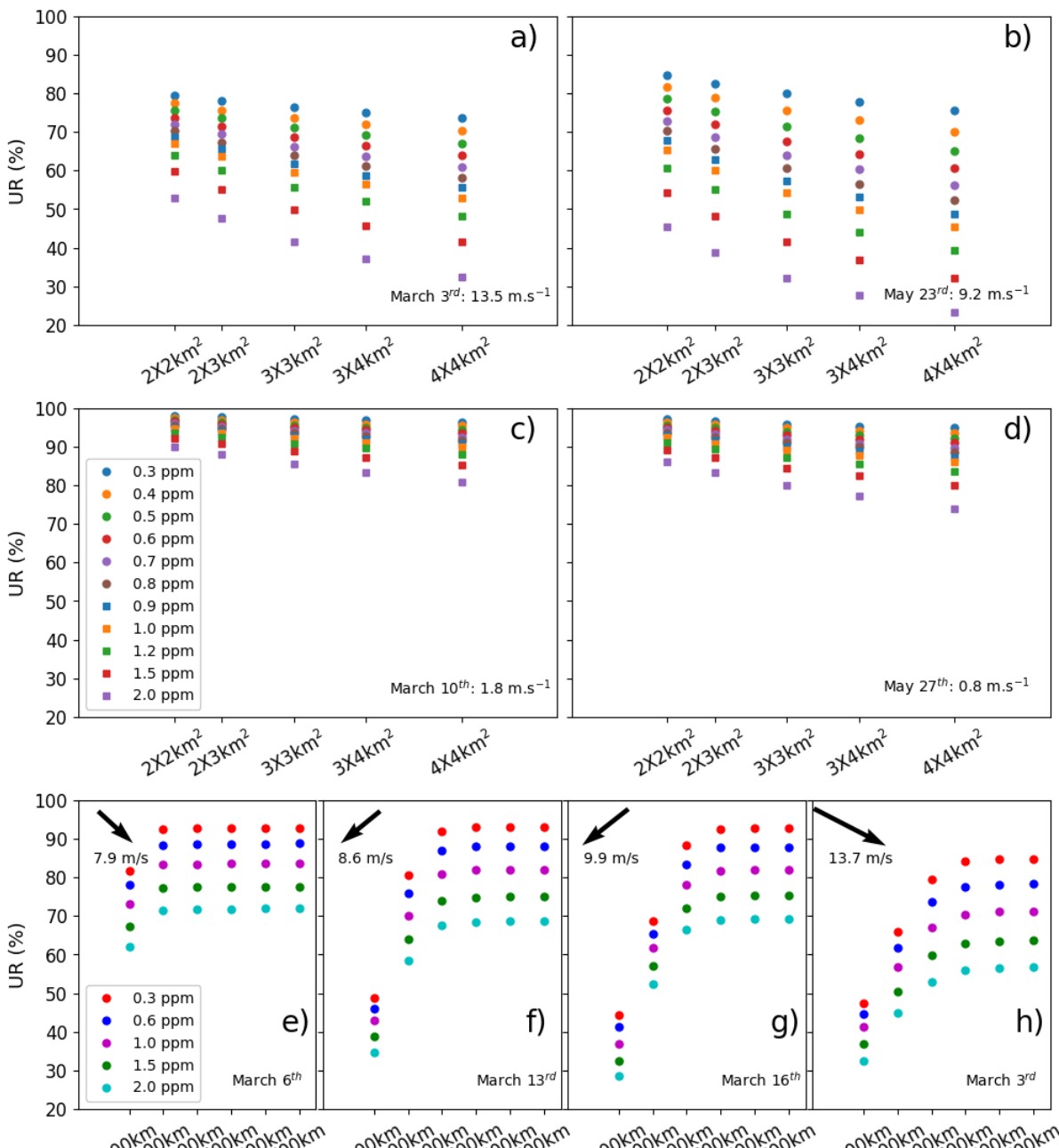

Figure 4: Uncertainty Reductions (UR) from a 50% prior uncertainty for the 6h-budgets of the Paris emissions. In panels (a)-(d), results are displayed for 4 different days characterized by different wind speeds, for different spatial resolutions of the satellite data (x-axis) and for different precisions (color markers). For the panels (a)-(d), results are generated by considering a swath of 300 km. In panels (e)-(h), results are displayed for 4 different days characterized by different wind speeds, for different swaths of the satellite data (x-axis), for different precisions (color markers) and for a resolution of 2 km by 2 km.

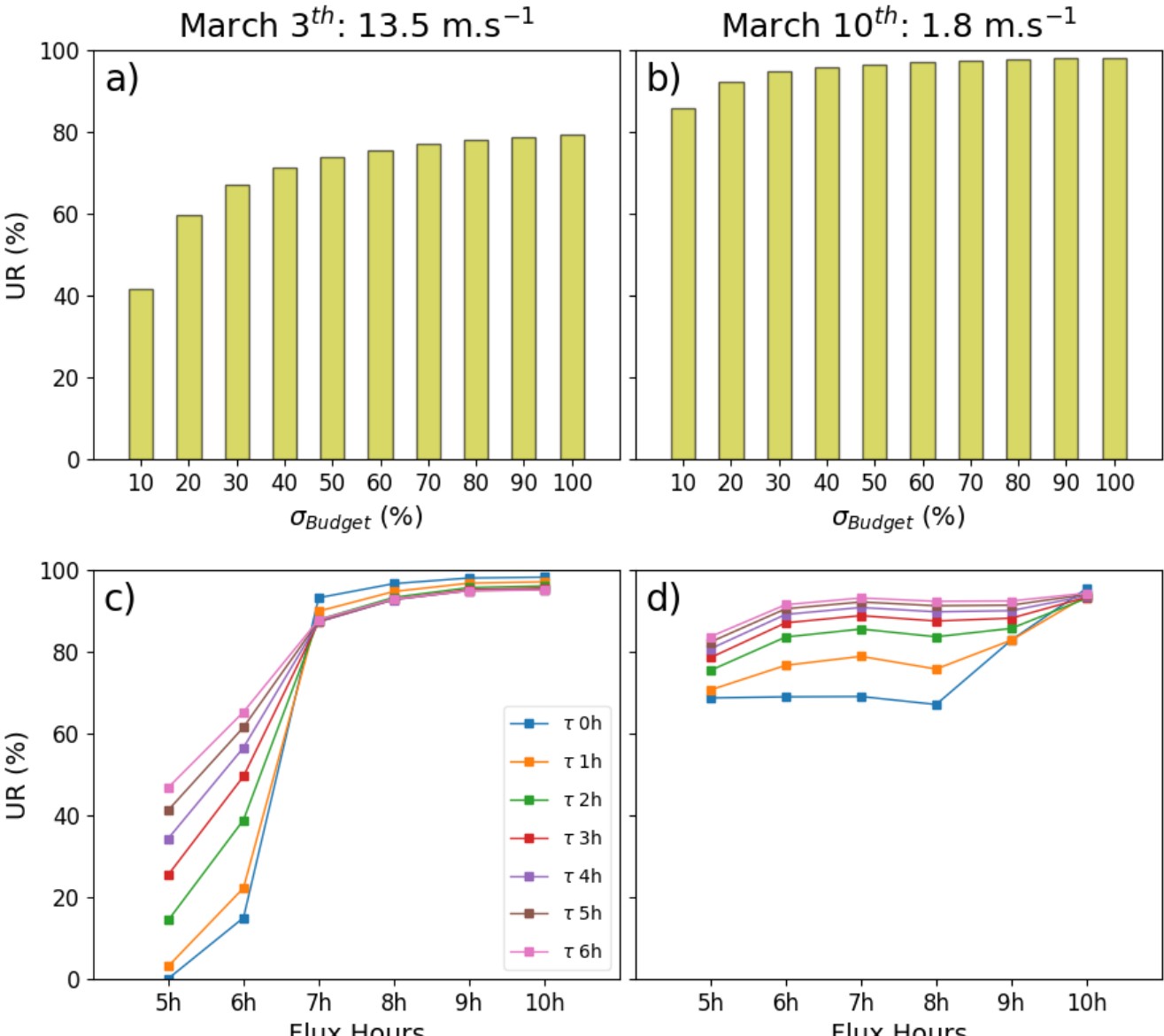

**Figure 5: Uncertainty Reductions (UR) as a function of the prior uncertainty (x-axis) for 6h-budgets of the Paris emissions (a,b). Correlations between the prior errors on hourly emissions have a temporal length of 3 hours (see section 2.4). Panels (c,d) show the UR for the hourly emissions between 5h and 11h (x-axis) for several temporal lengths defining the correlations between prior errors on hourly emissions (color dots), legend "τ 0h" being for an absence of such correlations. Prior uncertainties on 6h-budgets of Paris emissions are taken equal to 50% in panels (c,d). Columns represent 2 different inversion days: March 2016 the 3rd (strong wind) and March 2016 the 10th (low wind). All inversion results are computed with a retrieval resolution of 2 km × 2 km, a precision of 0.6 ppm and a swath of 300 km.**

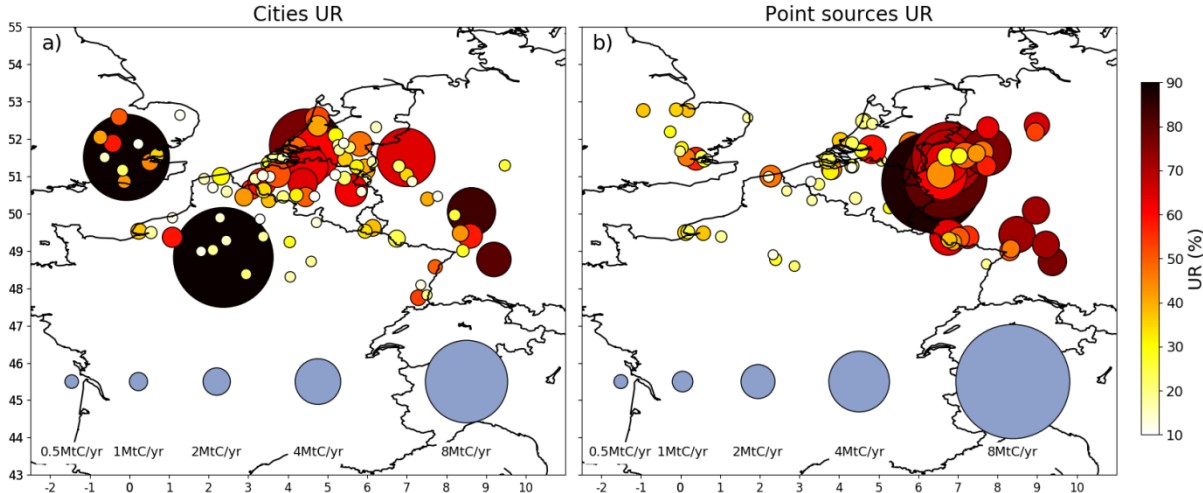

**Figure 6: Mean Uncertainty Reductions (UR) for some city clusters (a) and some point sources (b) across the 62 inversion results of the days of March and May 2016. The areas and colors of the disks represent the annual emissions (MtC·yr$^{-1}$) and the UR (%) respectively. The inversions are performed with a retrieval resolution of 2 km × 2 km, a precision of 0.6 ppm and a swath of 900 km. Prior uncertainties on 6h-budgets of clusters and point sources emissions are taken equal to 50% and prior error correlations have a temporal length of 3 hours.**

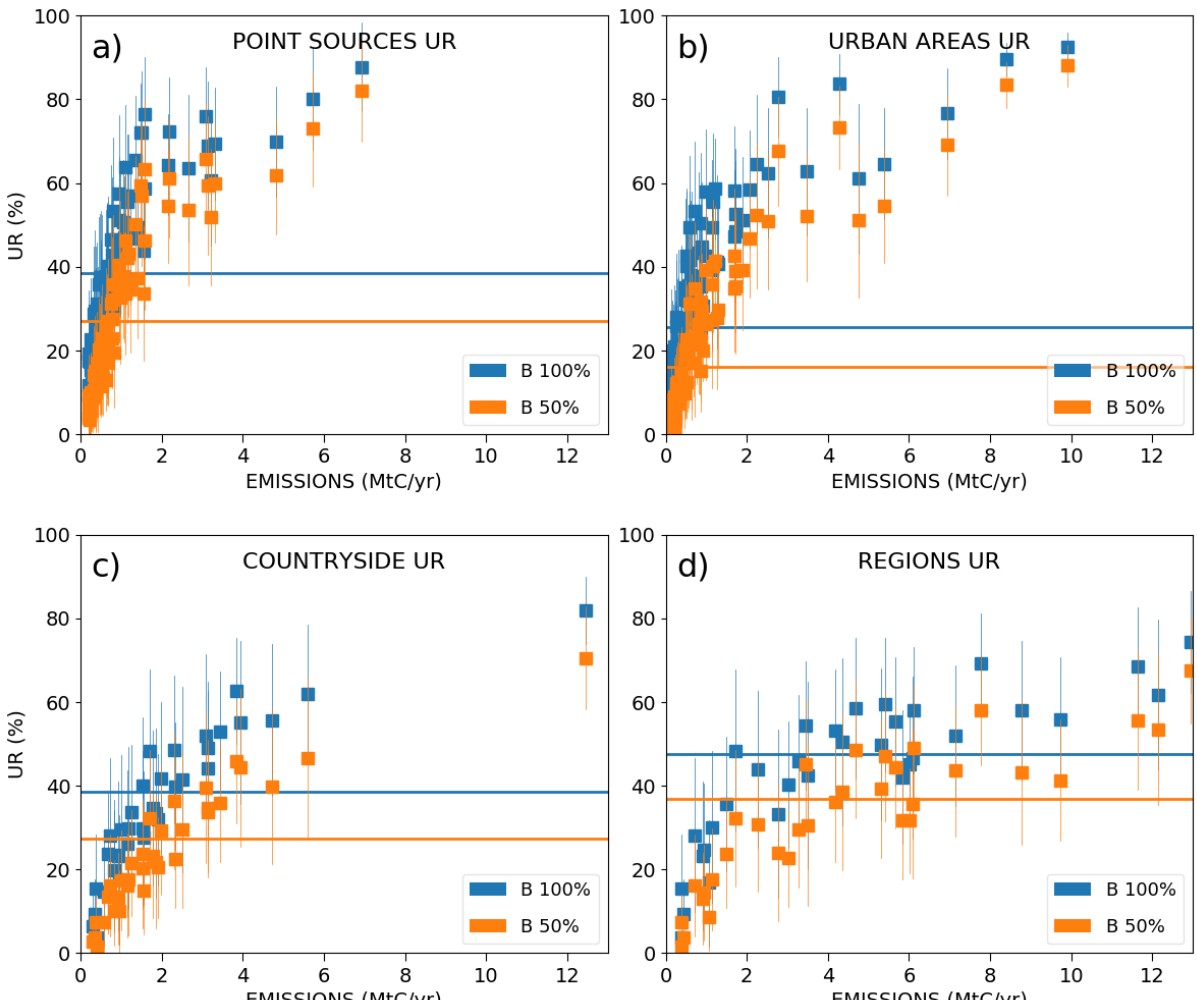

Figure 7: Mean values and standard deviations of the Uncertainty Reductions (UR) for the emissions of point sources (a), urban areas (b), countryside areas (c), and regions (d) across the 62 inversion results of the days of March and May 2016. The lines represent the averages of the temporal mean values across all sources of a given type. The emitting areas are chosen within the 2-km-resolution domain of the model so that they are covered by the satellite track in order to avoid swath effects. Results are given as a function of the annual emissions (x-axis) of the emitting areas. The inversions are performed with a retrieval resolution of 2 km × 2 km, a precision of 0.6 ppm and a swath of 900 km. Prior uncertainties on the 6h-budgets of the point sources, urban and countryside areas are taken equal to 50% (orange squares) and to 100% (blue squares). Prior uncertainties on the regional budgets are derived by aggregation of the prior uncertainties on their constituent emitting sources for both cases. Prior error correlations between hourly emissions have a temporal length of 3 hours.

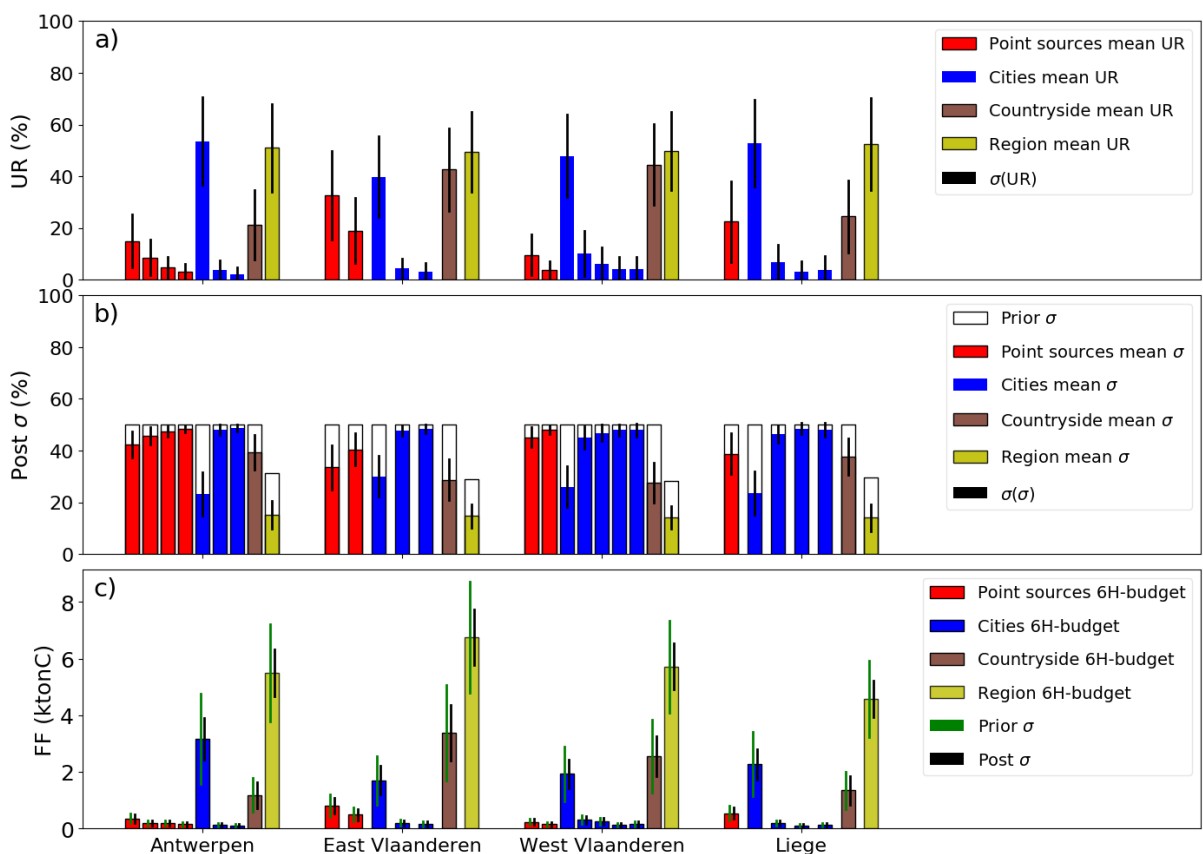

**Figure 8: (a) Mean values and standard deviations of the Uncertainty Reductions (UR) for the regions, point sources, urban and countryside areas constituting 4 Belgium regions. Averaging is performed across the 62 inversion results of the days of March and May 2016. (b) Mean values and standard deviations of the relative post uncertainty for each emitting area. Prior uncertainties are represented as well. Prior uncertainties in the regional budgets are derived by aggregation of the prior uncertainties of their**
**constituent emitting sources (mean value ~33%). Panel (c) shows the mean 6-hour budgets for each emitting area. The inversions are performed with a retrieval resolution of 2 km × 2 km, a precision of 0.6 ppm and a swath of 900 km. Prior uncertainties on the 6h-budgets of the point sources, urban and countryside areas are taken equal to 50% and prior error correlations between hourly emissions have a temporal length of 3 hours.**

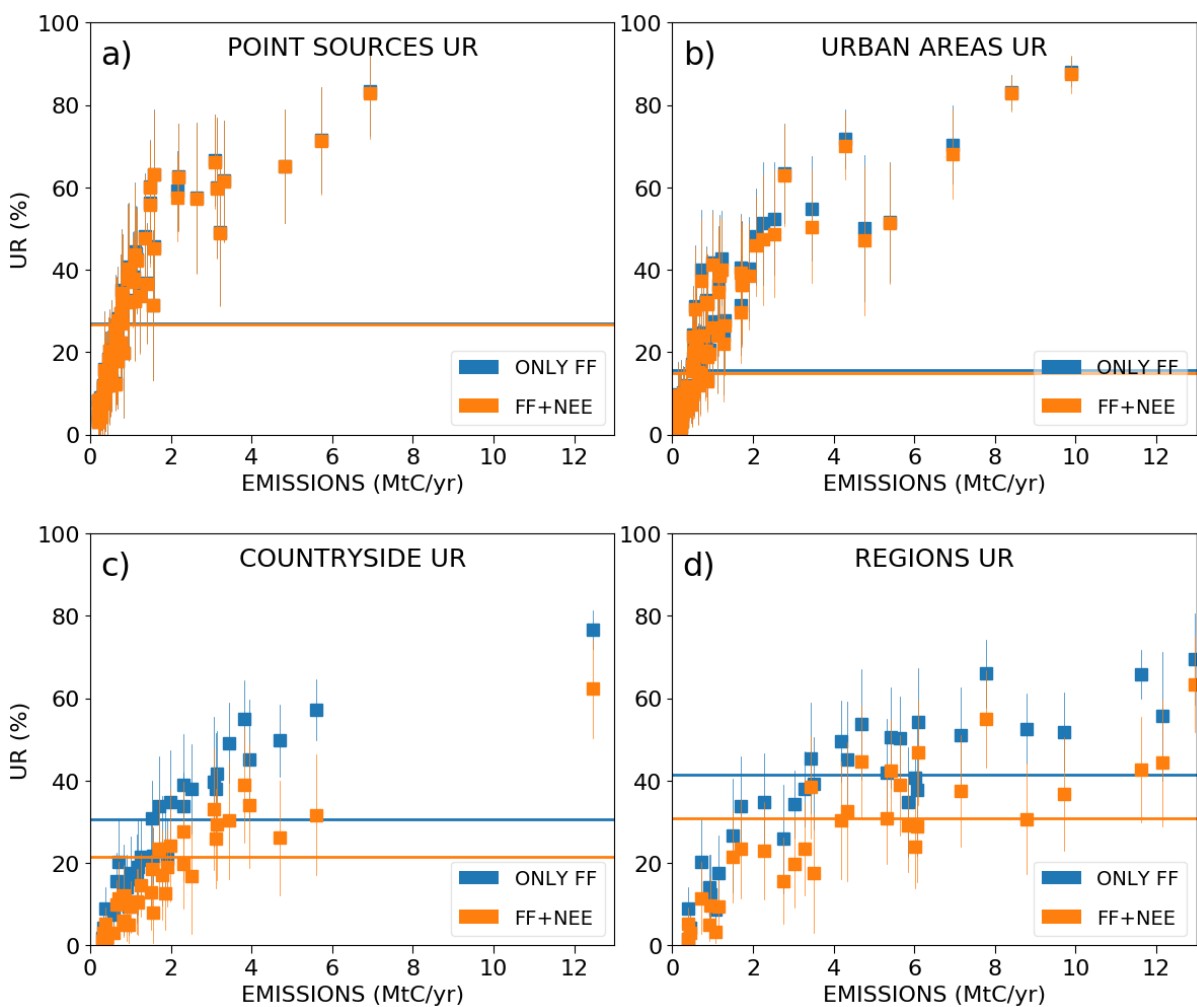

**Figure 9: Same as Fig. 7 but results are derived from inversions considering the anthropogenic emissions only (blue markers) and from inversions considering the natural fluxes as well (orange markers). Prior uncertainties on the 6h-budgets of the point sources, urban and countryside areas are taken equal to 50% and prior error correlations between hourly emissions have a temporal length of 3 hours. Prior uncertainties on the regional budgets are then derived by aggregation of the prior uncertainties of their constituent emitting sources (mean value ~33%).**

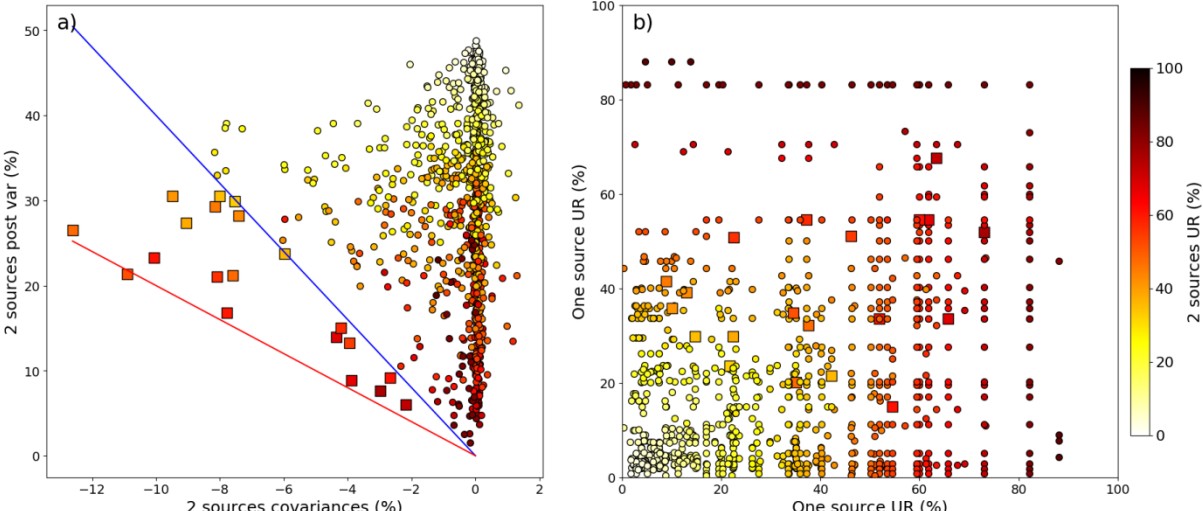

**Figure 10: a)** Post variances of the emission estimates for pairs of anthropogenic sources contained within a same region function of the covariances between the individual sources of the pairs. The colors of the markers correspond to the UR of the pairs. **b)** For a pair of anthropogenic sources contained within a same region, UR of one individual source function of the UR of the other individual source of the pair. The colors of the markers correspond to the UR of the pairs. The regions corresponding to this plot are contained within the 2 km-resolution area of the model. Results are derived with an instrumental resolution of 2 km × 2 km, a precision of 0.6 ppm and a swath of 900 km. Prior uncertainties on the 6h-budgets of the point sources, urban and countryside areas are taken equal to 50% and prior error correlations between hourly emissions have a temporal length of 3 hours. Pair of anthropogenic sources are made up of point sources and of urban and countryside areas.

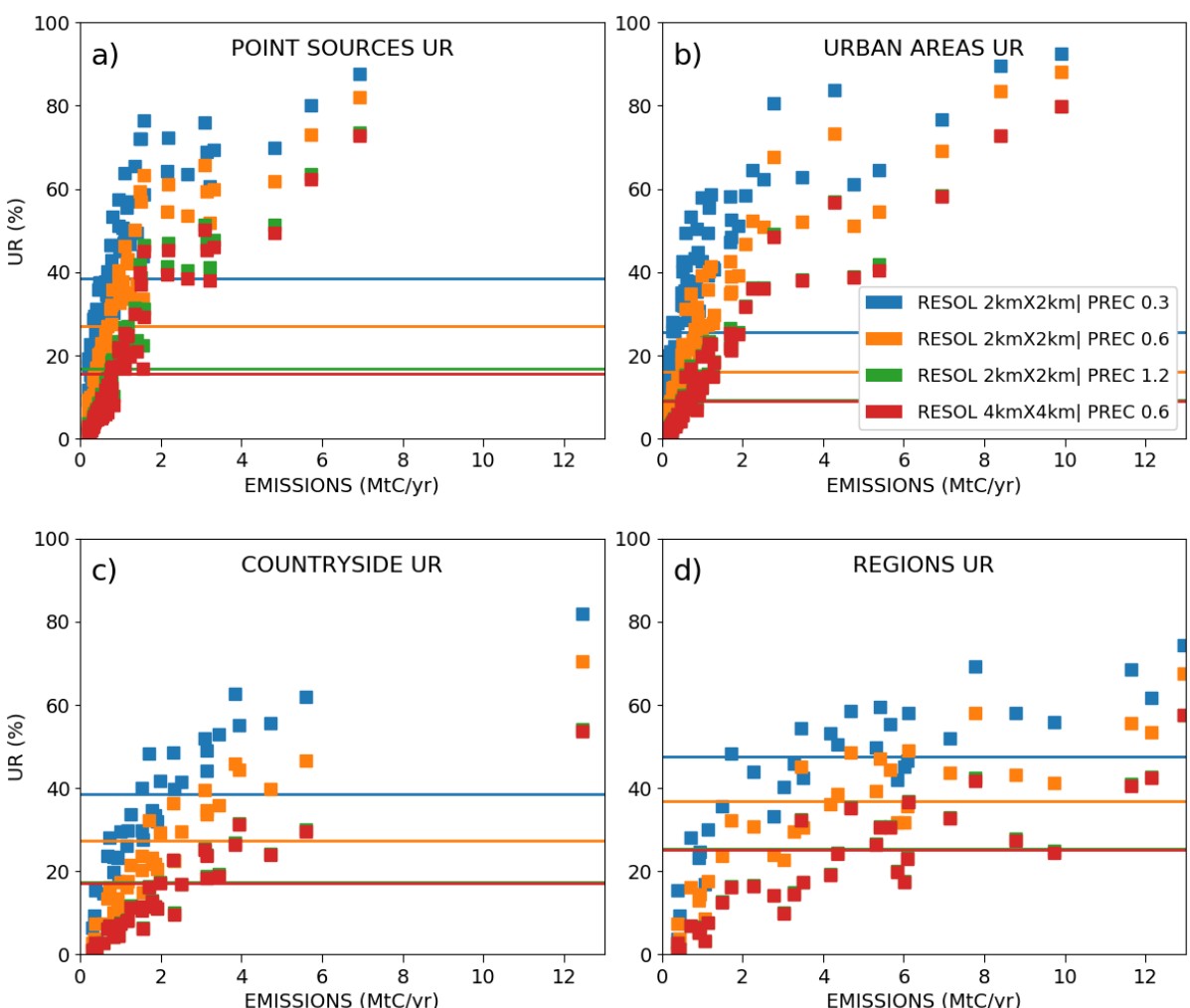

**Figure 11: Same as Fig. 7 but the inversions are performed with different retrieval resolutions and precisions (swath=900 km). Prior uncertainties in the 6h-budgets of the point sources, urban and countryside areas are taken equal to 50%.**

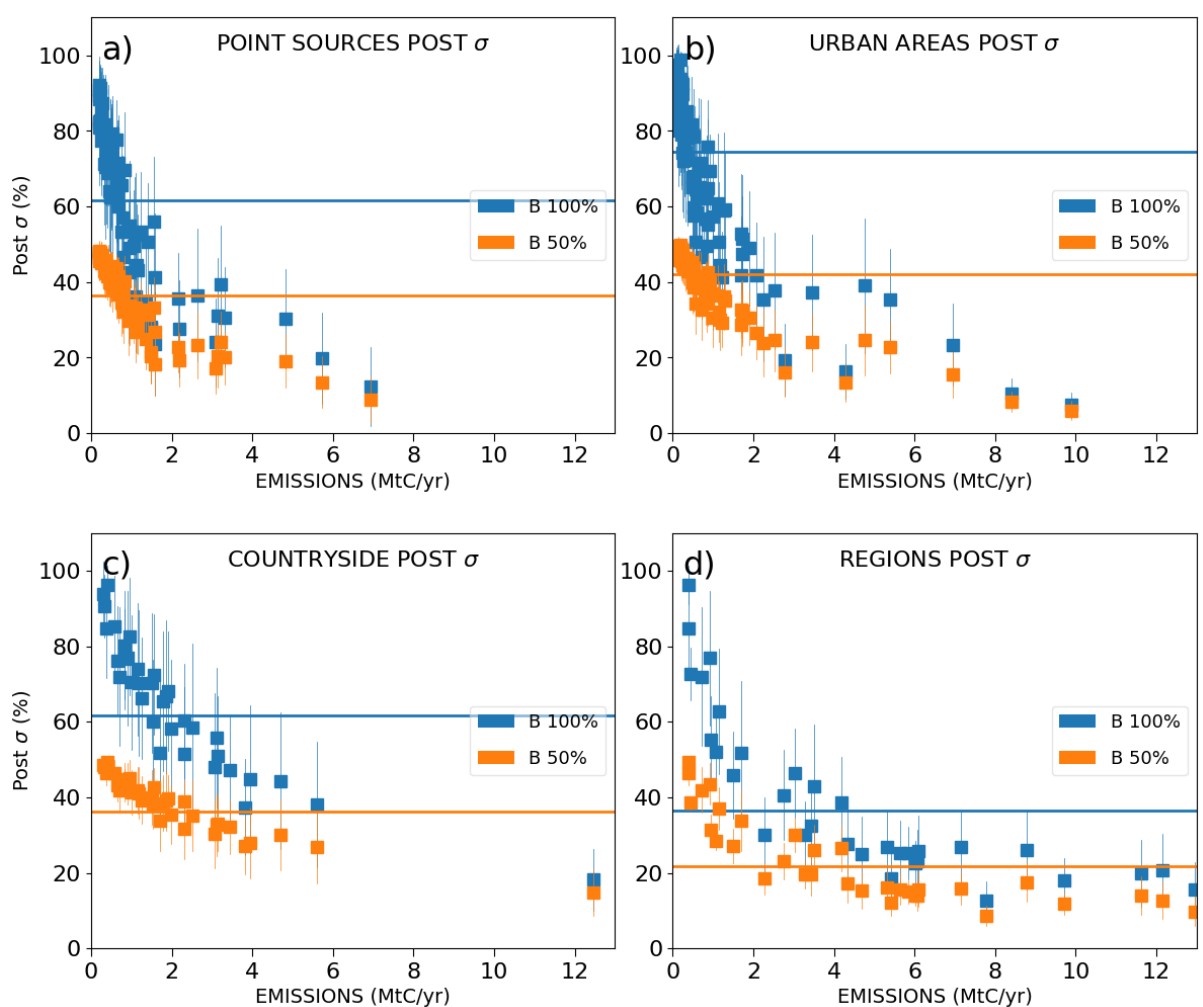

**Figure A1: Same as Fig. 7 but relative posterior uncertainties are shown instead UR.**

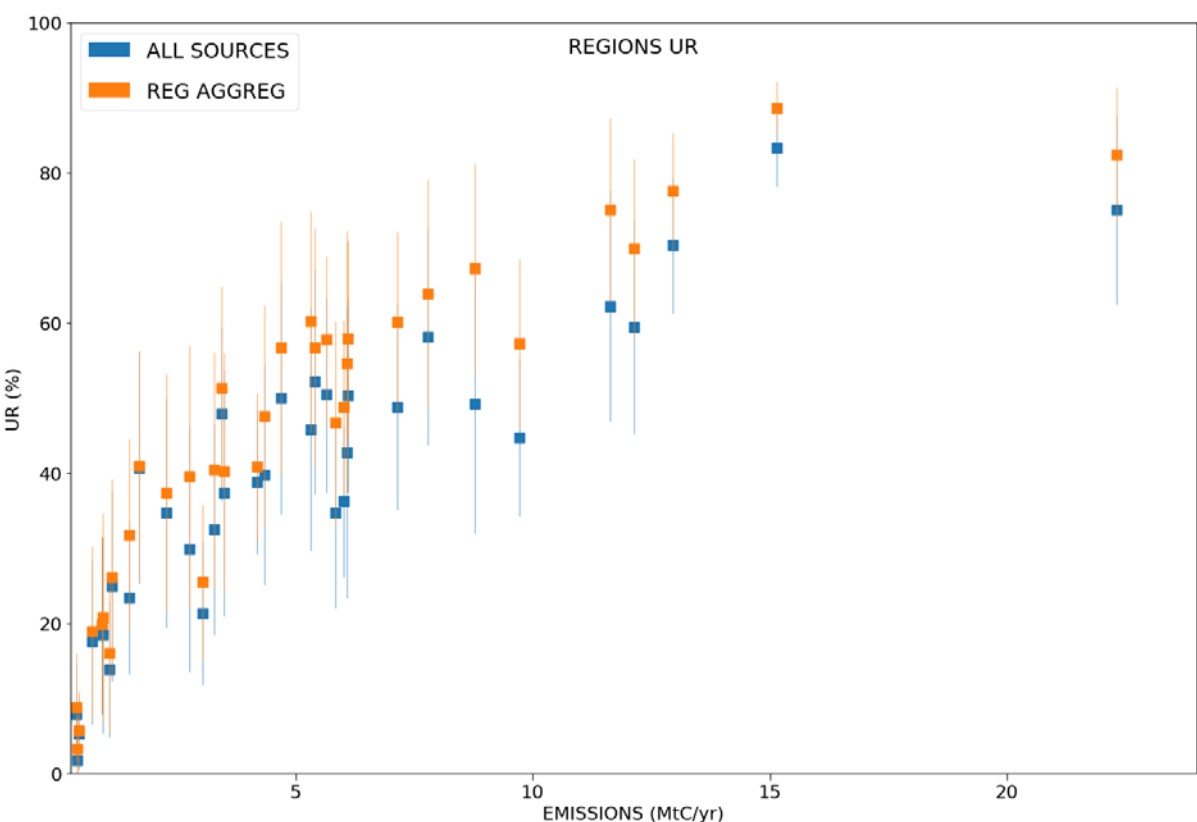

**Figure A2: Mean values of the Uncertainty Reductions (UR) of the emissions at the regional scale across the 62 inversion results of the days of March and May 2016. Results are given function of the annual emissions (x-axis). Regional emissions are inverted with (blue markers) and without (orange markers) considering an internal separation of the region into cities, point sources or countryside areas.**

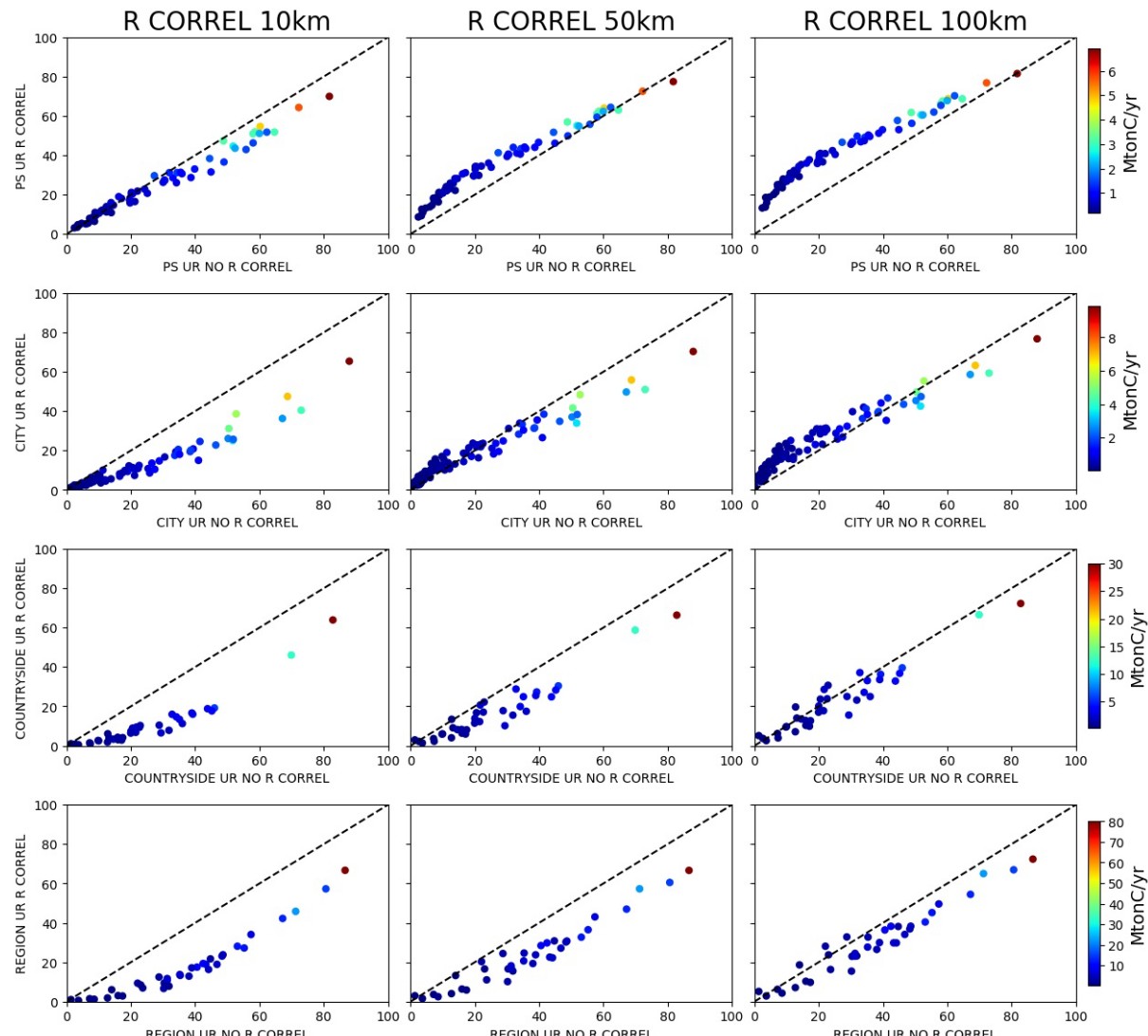

**Figure A3: Uncertainty reductions (UR) for the emissions of the point sources (1st row), the urban areas (2nd row), the countryside areas (3rd row) and the regions (4th row) when considering spatial correlations (Y-axis) or no correlations (X-axis) between the observation errors. The 1st, 2nd and 3rd column correspond to correlations with a spatial scale of 10, 50 and 100 km respectively. The color of the dots corresponds to the annual budgets of the sources (color bar on the right of the figure). The dashed line is the 1:1 line.**

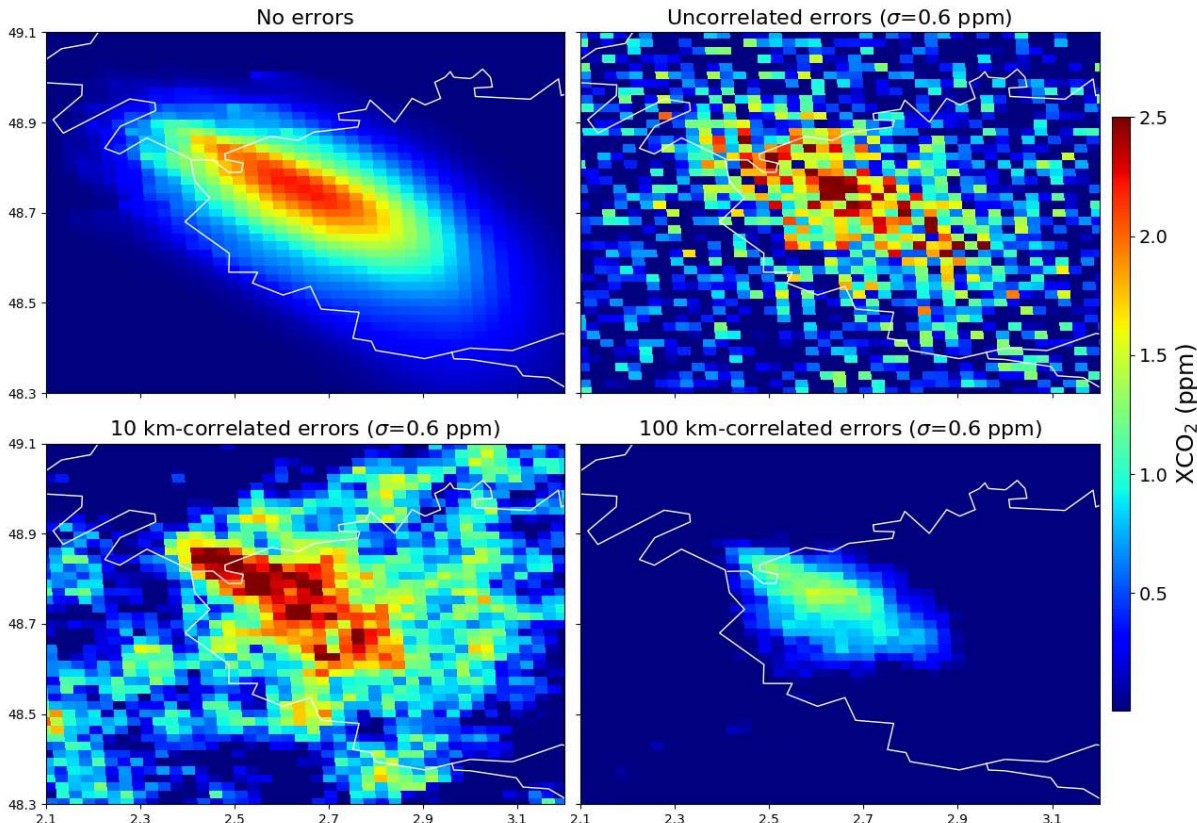

**Figure A4 : XCO$_2$ distributions produced by the emissions of the Paris area between 5:00 and 11:00 on March the 8$^{th}$ at 11:00: raw CHIMERE simulations (top left figure), raw CHIMERE simulations perturbed by an uncorrelated random noise of 0.6 ppm (top right figure), raw CHIMERE simulations perturbed by a multivariate normal distribution whose covariance matrix is characterized by an uncertainty of 0.6 ppm for the diagonal terms and a correlation length scale of 10 km (bottom left figure) and of 100 km (bottom right figure).**