# Peer review of "A local to national-scale inverse modeling system to assess the potential of spaceborne $CO_2$ measurements for the monitoring of anthropogenic emissions"

_Atmospheric Measurement Techniques, 2020_

## Referee Comment (RC1) · Anonymous Referee #1 · 24 Jun 2020

The authors do a good job of explaining a regional tool that can be used to evaluate observing systems. Much of this detail has already been covered in Broquet et al (2018), but they add some to the analysis presented there and extend to a more comprehensive state vector that includes the larger region, as well as exploring the impacts of satellite data precision and resolution on the inversions.

Everything as described is correct mathematically, and the results from the point of view of a linear least squares optimization are useful. I find the analysis related to the independence of various sources particularly interesting, as distinguishing from

neighboring sources is critical for the mission of CO2M.

As the authors highlight their exploration of the observational precision, it behooves me to point out that we have no reason to believe the assumption they make about independence of the errors in retrievals that are spatially near each other. In fact, work by Kulawik et al and Worden et al would suggest that the correlation length scales would be something more like 50km-100km for XCO2. The assumption that errors scale by sqrt(N) is particularly poor. I can appreciate that handling systematic errors in a classical uncertainty reduction framework is not straightforward, but handling correlated observation errors should be doable. This consideration is particularly important for small scale sources/sinks of the sort that the authors are claiming to constrain.

I think the paper is worthy of publication, but I do think that it will have more impact with this one extra factor considered.

I also recommend a bit more rigorous grammar and spelling check, as I noticed typographical errors and grammatical errors as I read the manuscript.

I don't have specific comments, as the presentation is straightforward, and the figures are self-explanatory.

—————————————————————

---

## Referee Comment (RC2) · Anonymous Referee #2 · 29 Jun 2020

**Review of Santaren et al (DOI:10.5194/amt-2020-138)**

The authors have done a series of inversions in an OSSE framework to explore the feasibility of a wide swath $CO_2$-sensing satellite mission such as CO2M to quantify the fossil fuel (FF) $CO_2$ emissions from individual cities, clusters of cities, and regions. They have relied on uncertainty reduction as the primary metric for evaluating their inversions. The work they have done is mathematically correct, and the results follow expectations from prior experience with inversions, i.e., they "make sense". However, there are three critical shortcomings, owing to which I cannot recommend publication unless they make the suggested changes. Since making these changes is likely to require a fair amount of additional work, and may change their conclusions, I am classifying them as "major" changes.

First, in any $CO_2$-based effort to quantify FF $CO_2$ emissions, the biggest confounding factor is the non-FF variation of $CO_2$. This can be due to biosphere fluxes over continents, or, in a regional study, due to inflow/outflow at the boundary. This has been a well-known problem both in the context of estimating FF $CO_2$ from in situ and satellite $CO_2$ measurements (e.g., https://doi.org/10.1002/2014GL059684 and https://doi.org/10.1029/2019JD030528 respectively). Biases in the assumed NEE – which is very likely in any biosphere model – **will** lead to biases in derived FF $CO_2$, which makes any uncertainty reduction irrelevant. And yet, in this set of OSSEs the authors skirt this very important issue, assuming the prior NEE to be unbiased. While this does not make the results wrong, it makes them less than useful for evaluating the potential of a CO2M-like mission, which will surely have to contend with unknown biospheric $CO_2$ fluxes.

Second, in the context of satellite $CO_2$ instruments, an additional complication is the data gap or poor data due to cloud cover, aerosol loading, and other factors. For example, for the currently flying OCO2 spectrometer, only a few percent of its footprints result in good quality $XCO_2$ retrievals. For the domain the authors have chosen (NW Europe with lots of urban centers), both cloud cover and aerosol loading are important limiting factors. Yet, the authors explicitly ignore this complication, "The cloud cover and the corresponding gaps in the spaceborne passive $XCO_2$ sampling are ignored" (L100). I understand that simulating realistic clouds and aerosols is difficult, but for an OSSE to be realistic, some attempt must be made. For example, the authors could have used the statistics of past cloud and aerosol data to introduce sampling gaps. Or, they could have taken the fraction of good quality retrievals among all footprints from an existing NIR $XCO_2$ instrument like OCO2 and then downsampled their footprints. The results presented here without considering realistic sampling gaps are mathematically correct, but not very useful for evaluating the capability of any real $CO_2$ mission.

Third, the authors use the posterior covariance matrix and uncertainty reduction as performance metrics for their inversions throughout the paper. Given the importance of uncertainty calculation to the work (as opposed to reduction of biases in their priors), I would like to see a more realistic specification of flux and data uncertainty. Currently they assume (i) uncorrelated retrieval errors (L193), which is unrealistic for the small footprint and dense sampling that they've given their satellite instrument, and (ii) no spatial correlation in their prior flux covariance **B** (L334), also unrealistic given the high spatial resolution of their fluxes. I do not understand why they need to make either simplification, since in a batch inversion (as opposed to an iterative approximation like EnKF or 4DVAR) one can actually specify off-diagonals in both **B** and **R**. The simplifications (i) and (ii) make their posterior covariance, and the conclusions based thereon, not very relevant for real-world inversions of CO2M-like satellite data.

A general comment I would like to make about the three issues I raise above is that in inversions of real satellite data, modelers often have to make simplifying choices to make the problem tractable. E.g., ignoring off-diagonal elements in **R** is pretty common, although more recent inversions try to at least account for correlations in **R** by error inflation, aggregating, or data thinning (e.g., https://doi.org/10.1029/2007GL030463, https://doi.org/10.5194/acp-13-8695-2013 and https://doi.org/10.5194/acp-19-9797-2019). Similarly, inversions with an iterative approximation like EnKF or 4DVAR often have an inexact posterior covariance due to computational limitations, while

many inversions ignore biases in satellite retrievals because it's still an open problem. However, making too many simplifications in a single OSSE, as the authors have done here (unbiased priors and retrievals, no sampling gaps, uncorrelated retrieval errors, no spatial error correlation in fluxes) makes the results inapplicable to any real-world satellite instrument. I will also point out that several co-authors of this manuscript have previously authored papers stressing the importance of these complicating factors (e.g, https://doi.org/10.1029/2007GL030463, https://doi.org/10.5194/amt-11-681-2018, and https://doi.org/10.1088/1748-9326/ab7835) and published inversions with far more realistic assumptions, which makes the current set of simplifications all the more surprising.

Beside these major issues, here are a few minor points that need correcting or clarifying:

1. L156: Delete "and vertical". The vertical resolution comes later.
2. L161: A high spatial resolution (~2x2 $km^2$) implies higher temporal resolution as well. If the driving winds are 3-hourly, what provides high frequency variation in the CHIMERE winds?
3. L175: The gradients in column $CO_2$ due to the top 30% of the atmosphere would be small, agreed, but how large are they? Signals in column $CO_2$ are deceptively small, so terms that seem to be negligible are not always negligible.
4. L180: Switch 92.8° and 705 km.
5. L227: Since the quantity directly estimated is the FF $CO_2$ emission between 5 and 11 local time, to estimate the total emissions one would need an accurate diurnal cycle. What is the uncertainty in the diurnal cycle of FF $CO_2$ emissions?
6. Section 2.2.2 and elsewhere: The word "controlled" keeps confusing me. Do you mean "estimated", as in part of the control vector? Or do you mean "controlled" as in kept in control, static, not changed? I suspect you mean the former, but "controlled" in English can also signify "not allowed to change". I'd suggest using the word "estimated" or "optimized" if you mean the former.
7. L280: Is this an over-determined problem? Then that's not very common in flux inversions, and is likely due to the unrealistic correlations in **R** and **B** (one of my major concerns).
8. L290: Is random noise added to **y**, consistent with **R**?
9. L304: Typo, change $XO_2$ → $XCO_2$
10. L348: With the assumptions detailed here at the grid scale, what is the uncertainty on the (say) annual total or seasonal total NEE and FF? Aggregate numbers are easier to make sense of than grid-scale specifications.
11. L399: "Figures 2i" likely means all the subplots of Figure 2. In that case, just say "Figures 2", no need to add the "i".
12. L437: Speed is one aspect of the wind, direction is the other. Since wind direction determines how well plumes present themselves to a satellite that is going one way, uncertainty in wind direction must be considered as well as speed uncertainty. Was that done here?
13. L477: Remove "uncertainty", **B** is just the prior covariance matrix.
14. L519: Again, I'd like to see the uncertainties on aggregated fluxes, such as annual totals.
15. L560: In Figure 7, why do larger emissions have smaller uncertainty reductions?
16. L582: "… *and thus by the variability of these fluxes, during the month of May*". This only matters because the uncertainty on the NEE is larger in May, right? Because this metric/score does not care about the actual prior NEE.
17. L794: "*Efforts have been made to limit the amplitude of such errors in the concept of the new CO2M mission. Our new inversion framework allows accounting for a realistic simulation of the observation sampling and errors.*" I disagree with this statement in the context of this paper, especially the part about a realistic simulation, because of the three major points raised above.

---

## Author Comment (AC1) · 16 Oct 2020

We thank both reviewers for their comments on our manuscript. Thanks to them, this manuscript now conducts a discussion on the interesting topic of the spatial correlation in the observation errors, based on new tests described in Appendix A of the new manuscript. We provide in supplement the detailed responses to all of their comments and the version of the new manuscript where the changes are highlighted.

Please also note the supplement to this comment:

https://amt.copernicus.org/preprints/amt-2020-138/amt-2020-138-AC1-supplement.pdf

---

## Author Response (AR1)

We thank both reviewers for their comments on our manuscript. Thanks to them, this manuscript now conducts a discussion on the interesting topic of the spatial correlation in the observation errors, based on new tests described in Appendix A of the new manuscript. We provide below detailed responses to all of their comments and the version of the new manuscript where the changes are highlighted.

**Response to the anonymous Referee #1**

The authors do a good job of explaining a regional tool that can be used to evaluate observing systems. Much of this detail has already been covered in Broquet et al (2018), but they add some to the analysis presented there and extend to a more comprehensive state vector that includes the larger region, as well as exploring the impacts of satellite data precision and resolution on the inversions.

Everything as described is correct mathematically, and the results from the point of view of a linear least squares optimization are useful. I find the analysis related to the independence of various sources particularly interesting, as distinguishing from neighboring sources is critical for the mission of CO2M.

We thank the reviewer for these positive general statements.

As the authors highlight their exploration of the observational precision, it behooves me to point out that we have no reason to believe the assumption they make about independence of the errors in retrievals that are spatially near each other. In fact, work by Kulawik et al. (2020) and Worden et al. (2017) would suggest that the correlation length scales would be something more like 50km-100km for XCO2. The assumption that errors scale by sqrt(N) is particularly poor.

We made explicit in Section 2.1.2 that we purposely focus on the impact of the instrumental noise and we now clarify it in the introduction of our revised manuscript while mentioning the exploration of the topic of the correlations in the Appendix A:

*In terms of errors in the $XCO_2$ data, the analysis focuses on random errors due to the instrumental noise that have no spatial correlations (even though the topic is explored in Appendix A).*

We assume that errors in XCO2 due to this instrumental noise bear no spatial correlations, following, e.g., Buchwitz et al. (2013). The analysis of current OCO-2 data at high spatial resolution demonstrates that the uncorrelated noise is a significant fraction of the total error on XCO2 individual data (e.g. Reuter et al., 2019; Zheng et al. 2020).

Studying the correlated errors from the radiative transfer inverse modelling was out of the scope of our study.

I can appreciate that handling systematic errors in a classical uncertainty reduction framework is not straightforward,

Systematic errors can be accounted for in a classical uncertainty reduction framework, for instance based on a Monte Carlo approach (Broquet et al., 2018). However, assigning them remains difficult because:

- they are not described in the uncertainties calculated by existing retrieval schemes, in contrast to the retrieval noise caused by instrument noise. The publications cited above characterize them empirically based on the statistics of the retrieval small-scale variability (Worden et al., 2017) or on the statistics of the difference to reference retrievals that are themselves empirically related to WMO standards (Kulawik et al., 2019a,b).
- they depend on the specific measurement configurations and on the evolving skill of radiative transfer models and of empirical bias-correction systems, so that conclusions from existing missions may hardly apply to future ones.

but handling correlated observation errors should be doable.

Including correlations in the **R** matrix of our inversion framework is definitely doable and we have conducted some tests with such correlations to support our discussion on this topic. In order to lighten the computations associated with the inversion of such a matrix when using a 900 km swath, we have actually tested it when considering 4 km resolution pixels rather than 2 km resolution ones.

A traditional way to model spatial correlations is to assume that they are isotropic and homogeneous in terms of spatial scale (as probably suggested by the reviewer when speaking about 50 to 100 km scale correlations). We have modeled such correlations using exponentially decaying functions exp(-d/D) of the distances d between two observation pixels, and tested their inclusion in **R** with a 1-sigma uncertainty of 0.3 ppm for individual $XCO_2$ data and with D=10, 50 or 100 km (assuming the correlations apply to the total uncertainty in individual data i.e. not splitting **R** into a random noise component and a component with spatial correlations). These tests are now described in Appendix A where the following Figure A3 is also provided. These results and this topic are now discussed in Section 4, with the following main messages:

- such isotropic correlations should make the errors much easier to filter by the inversion than actual spatial patterns of the systematic errors that can follow the same atmospheric dynamics as the signature of the targeted fluxes
- the situation is made even more optimistic here (like in most of studies with atmospheric inversion OSSEs) since our inversions are perfectly informed about the statistics of the observation errors (and in particular about their spatial scale)
- as expected, the results indicate that when introducing correlations with small spatial scales, the posterior uncertainties increase since these correlations yield larger budget of observation errors at the scale of the signatures of the targeted fluxes. Conversely, introducing correlations helps the inversion separate the observation error patterns from these signatures, and large correlation spatial scales lead to a decrease of the posterior uncertainties. The worst case corresponds to correlation scales that are function of the scale of the signatures of the targeted fluxes. It varies depending on whether we analyze results for point sources, cities or widespread emissions across regions. This is why (i) results are poorer with D=10 km than with no correlations, for all type of sources (ii) results become generally better with spatial correlation than without for point sources and cities respectively for D≥10 km and D≥50km (iv) the

inversion of the largest of a given category of sources (in terms of amplitude) are generally more negatively or less positively impacted by the spatial correlations (iii) results are systematically worse with D= 10 to 100 km than with no correlations, for countryside emissions. Results for regional budgets mix all these behaviors in a complex way.

- These results and conclusions should be further investigated in future theoretical studies (to properly analyze the characterization of the spatial scale of the signature of the targeted fluxes) but this would fall out of the scope of this paper since (i) we want to keep the focus on the impact of random instrumental noise (ii) we believe that such a simple structure of the error correlations and their optimal learning by the inversion should be interpreted carefully to avoid misleading conclusions regarding the impact of actual systematic errors in the inversions.

Here is the corresponding text added in section 4:

*Our new inversion framework allows accounting for a realistic simulation of the observation sampling and errors. Nevertheless, generating simulations of the systematic errors from the retrieval of $XCO_2$ data that are suitable for the purpose of our study would have been difficult. Systematic errors are not described in the uncertainties computed by existing retrieval schemes. Furthermore, they depend on specific measurement configurations and on the evolving skill of radiative transfer inverse models and of empirical bias-correction systems, so that their characterization based on diagnostics with existing missions may hardly apply to future ones. Simulating realistic patterns of cloud cover consistent with the meteorology for the different test cases investigated would have also been challenging. Finally, this study focuses on other parameters to allow exploring the sensitivity of the inversions to these parameters in depth.*

*In order to raise insights into the impact of errors with spatial patterns such as model and systematic errors, we have conducted experiments where spatial correlations are included in the observation error (in the **R** matrix). We have tested isotropic and homogeneous spatial correlations exponentially decaying with distance, using various correlation lengths. The experiments and results are described in Appendix A since they are out of the scope of this study. The results indicate that including correlations in the observation errors tends to increase the budget of observation errors and thus to increase the posterior uncertainties in the flux estimates as long as the correlation length scale does not exceed that of the signature of the fluxes in the $XCO_2$ images. However, including correlations in the observation errors also tends to increase the ability to distinguish between the patterns of the observation error and of the signatures of the fluxes, and thus to decrease the posterior uncertainties, so that for large spatial correlation lengths, increasing the correlation length leads to a decrease of the posterior uncertainties. In our tests, the worst situation for the monitoring of the emissions in the study area corresponds to ~10 km correlation length scales. These results should be interpreted cautiously since the spatial patterns of the model and systematic errors are more complex than this traditional but simple modeling of spatial correlations and since in these tests, the inversion system if perfectly informed about the statistics of the observation error. Future studies will integrate more realistic simulations of observation sampling and errors from different concepts of spaceborne imagery, based on radiative transfer inverse modeling applied to realistic fields of surface and atmospheric conditions and instrumental specifications.*

[Figure]

*Figure A3: Uncertainty reductions (UR) for the emissions of the point sources (1st row), the urban areas (2nd row), the countryside areas (3rd row) and the regions (4th row) when considering spatial correlations (Y-axis) or no correlations (X-axis) between the observation errors. The 1st, 2nd and 3rd column correspond to correlations with a spatial scale of 10, 50 and 100 km respectively. The color of the dots corresponds to the annual budgets of the sources (color bar on the right of the figure). The dashed line is the 1:1 line.*

This consideration is particularly important for small scale sources/sinks of the sort that the authors are claiming to constrain.

The dependence of the impact of such correlations on the type of sources is definitely an important aspect of the problem as highlighted above.

I think the paper is worthy of publication, but I do think that it will have more impact with this one extra factor considered.

As explained above, we have tried to consider this extra factor in the new version, while respecting the coherence of the study (by providing the details of the tests of sensitivity to the

spatial correlations in **R** in Appendix A) and acknowledging the unknown evolving nature of systematic errors.

I also recommend a bit more rigorous grammar and spelling check, as I noticed typographical errors and grammatical errors as I read the manuscript.

We have carefully proofread the text to significantly improve its grammar and spelling (see track changes in the revised manuscript).

I don't have specific comments, as the presentation is straightforward, and the figures are self-explanatory.

We thank the reviewer again for such a positive assessment.

**Response to the Anonymous Referee #2**

The authors have done a series of inversions in an OSSE framework to explore the feasibility of a wide swath $CO_2$-sensing satellite mission such as CO2M to quantify the fossil fuel (FF) $CO_2$ emissions from individual cities, clusters of cities, and regions. They have relied on uncertainty reduction as the primary metric for evaluating their inversions. The work they have done is mathematically correct, and the results follow expectations from prior experience with inversions, i.e., they "make sense".

We thank the reviewer for his comments and for raising detailed discussions on our inversion set-up.

However, there are three critical shortcomings, owing to which I cannot recommend publication unless they make the suggested changes. Since making these changes is likely to require a fair amount of additional work, and may change their conclusions, I am classifying them as "major" changes.

As detailed below, the first "shortcoming" raised by the reviewer does not apply to the inversions that we have presented. Furthermore, we do not agree that suggestions of changes related to the "shortcoming #2 and #3" should be required. We disagree with many details of these suggestions, and we view the discussions on the corresponding topics as incentive for further investigations, while the set of original experiments is already significant, and while the manuscript is already long, analyzing many aspects of the inversion and satellite CO2 data assimilation problems. However, since the first part of the third suggestion is also raised by the first reviewer, we provide some analysis in Appendix A and some discussions in the manuscript to address the corresponding topic.

First, in any CO2-based effort to quantify FF CO2 emissions, the biggest confounding factor is the nonFF variation of CO2. This can be due to biosphere fluxes over continents, or, in a regional study, due to inflow/outflow at the boundary. This has been a well-known problem both in the context of estimating FF CO2 from in situ and satellite CO2 measurements (e.g., https://doi.org/10.1002/2014GL059684 and https://doi.org/10.1029/2019JD030528 respectively). Biases in the assumed NEE – which is very likely in any biosphere model – will lead to biases in derived FF CO2, which makes any uncertainty reduction irrelevant. And yet,

in this set of OSSEs the authors skirt this very important issue, assuming the prior NEE to be unbiased.

This first "major" comment is misplaced since our control vector does include both NEE fluxes at hourly and regional scale and the boundary conditions so that uncertainties in these fluxes and conditions are clearly accounted for. We describe the prior NEE errors in terms of random time-correlated uncertainties rather than in terms of biases (to use the reviewer's words), but:
- this is the usual statistical representation of uncertainties is NEE in inversions, in particular in those targeting NEE estimates (Rayner et al., 2019)
- we do not see how this distinction would change the conclusions of the paper

Furthermore, specific tests are conducted to analyze the impact of uncertainties in the NEE in section 3.3.2 which also discusses this "confounding factor" based on the correlations between the uncertainties in these fluxes and the anthropogenic emissions.

However, we take the opportunity of such a comment to revise section 2.4 to better clarify the set-up of uncertainties in the NEE and boundary conditions in the **B** matrix.

Here are the corresponding sentences added or modified in section 2.4. (see track changes in the revised manuscript):

- *Prior estimates of the boundary conditions for regional inversions are usually interpolated from large scale analysis or inversions. Such products can bear significant large-scale errors at the boundaries of Europe (Monteil et al., 2018). We reflect it by setting in **B** the standard deviation of the prior uncertainty in the scaling factor for the boundary conditions (see below).*
- *Finally, we use 1% for the STD of the prior uncertainty in the scaling factor associated to the boundary conditions (i.e. typically an uncertainty of ~4 ppm in the average boundary conditions). This value is quite pessimistic, but some tests in which this value was varied (not shown) demonstrate a very weak sensitivity of the results for the fluxes to this parameter.*
- *Despite the differences between the temporal variations of the hourly emissions from one control area to the other, or between natural and anthropogenic fluxes in $H_{distr}$, these STD show a negligible variation of less than 1% and when considering the reference set-up for **B**, $\sigma_{hour}$~65%.*

While this does not make the results wrong, it makes them less than useful for evaluating the potential of a CO2M-like mission, which will surely have to contend with unknown biospheric CO2 fluxes.

Again, we explicitly account for uncertainties in the NEE in our inversions and we have a result and analysis section dedicated to the topic.

Second, in the context of satellite CO2 instruments, an additional complication is the data gap or poor data due to cloud cover, aerosol loading, and other factors. For example, for the currently flying OCO2 spectrometer, only a few percent of its footprints result in good quality XCO2 retrievals. For the domain the authors have chosen (NW Europe with lots of urban centers), both cloud cover and aerosol loading are important limiting factors. Yet, the authors explicitly ignore this complication, "The cloud cover and the corresponding gaps in the

spaceborne passive XCO2 sampling are ignored" (L100). I understand that simulating realistic clouds and aerosols is difficult, but for an OSSE to be realistic, some attempt must be made. For example, the authors could have used the statistics of past cloud and aerosol data to introduce sampling gaps. Or, they could have taken the fraction of good quality retrievals among all footprints from an existing NIR XCO2 instrument like OCO2 and then downsampled their footprints. The results presented here without considering realistic sampling gaps are mathematically correct, but not very useful for evaluating the capability of any real CO2 mission.

Accounting for cloud cover would be important if we had to analyze results for a specific mission with a fixed spatial and temporal sampling, and over a long period of time, in order to highlight the frequency of its cloud free observations and potentially some asset of using a higher spatial resolution for the observations (as discussed in section 4). However, our study mainly focuses on the sensitivity of the inversion to other parameters. Adding the impact cloud cover, with its short-term chaotic component, in the experiments would not help raise such sensitivity curves and understand them. Furthermore, characterizing the sensitivity to cloud cover itself for individual images of plumes is not straightforward since it depends on how cloud patterns overlap with the plumes or not, especially if the set of cloud cover maps used for the study is limited. Of important note is that the impact of testing various cloud cover for the inversion of the emissions from a given city was analyzed in our previous study (Broquet et al., 2018).

From a technical point of view, the reviewer's suggestion is challenging. The point is not just about having a realistic percentage of pixels removed from the image due to cloud cover, but also to have realistic pattern of gaps that are consistent with the meteorology. In particular, a uniform spread of cloudy pixels would necessarily impact the observation of a given plume in a partial way, while structured cloud coverage could either leave intact or completely hide a given plume. A way to tackle the problem is generally to simulate XCO2 data based on real earth observation data. For example, Buchwitz et al. (2013) simulated the sampling by the proposed CarbonSat mission using MODIS data. This was relatively straightforward since the CarbonSat orbit was taken as that of Terra, since its swath was narrower and since its spatial resolution was coarser than that of MODIS data. However, here, we simulate theoretical observations every day during several months at various spatial resolutions. Building assumptions and processing MODIS data to achieve such a simulation would be a study in itself, and is clearly out of the scope of the present paper. We thus disagree with the reviewer's suggestion to add an analysis of the impact of cloud cover on the results.

We now clarify this position in the manuscript (section 4):

*Simulating realistic patterns of cloud cover consistent with the meteorology for the different test cases investigated would have also been challenging. Finally, this study focuses on other parameters to allow for exploring the sensitivity of the inversions to these parameters in depth.*

Third, the authors use the posterior covariance matrix and uncertainty reduction as performance metrics for their inversions throughout the paper.

This is not not specific to this paper at all. It is a standard way to diagnose the skill of inversions (Rayner et al., 2019).

Given the importance of uncertainty calculation to the work (as opposed to reduction of biases in their priors),

There is no opposition: in the optimal estimation framework, the Kalman gain matrix that controls the random uncertainty reduction (together with the observation operator) also reduces biases in the prior. The mathematical demonstration is trivial.

I would like to see a more realistic specification of flux and data uncertainty. Currently they assume (i) uncorrelated retrieval errors (L193), which is unrealistic for the small footprint and dense sampling that they've given their satellite instrument,

We do not agree with this point since we explicitly focus on the random measurement errors from instrumental noise. Such correlations arise in the systematic errors that have been ignored in this study. As written to reviewer 1:

We made explicit in Section 2.1.2 that we purposely focus on the impact of the instrumental noise and we now clarify it in the introduction of our revised manuscript while mentioning the exploration of the topic of the correlations in the Appendix A:

*In terms of errors in the $XCO_2$ data, the analysis focuses on random errors due to the instrumental noise that have no spatial correlations (even though the topic is explored in Appendix A).*

We assume that errors in XCO2 due to this instrumental noise bear no spatial correlations, following, e.g., Buchwitz et al. (2013). The analysis of current OCO-2 data at high spatial resolution demonstrates that the uncorrelated noise is a significant fraction of the total error on XCO2 individual data (e.g. Reuter et al., 2019; Zheng et al. 2020).

Studying the correlated errors from the radiative transfer inverse modelling was out of the scope of our study.

Systematic errors can be accounted for in a classical uncertainty reduction framework, for instance based on a Monte Carlo approach (Broquet et al., 2018). However, assigning them remains difficult because:

- they are not described in the uncertainties calculated by existing retrieval schemes, in contrast to the retrieval noise caused by instrument noise. The publications cited above characterize them empirically based on the statistics of the retrieval small-scale variability (Worden et al., 2017) or on the statistics of the difference to reference retrievals that are themselves empirically related to WMO standards (Kulawik et al., 2019a,b).
- they depend on the specific measurement configurations and on the evolving skill of radiative transfer models and of empirical bias-correction systems, so that conclusions from existing missions may hardly apply to future ones.

However, including correlations in the **R** matrix of our inversion framework is definitely doable and we have conducted some tests with such correlation to support our discussion on this topic. In order to lighten the computations associated with the inversion of such a matrix when using a 900 km swath, we have actually tested it when considering 4 km resolution pixels rather than 2 km resolution ones.

A traditional way to model spatial correlations is to assume that they are isotropic and homogeneous in terms of spatial scale (as probably suggested by the reviewer when speaking about 50 to 100 km scale correlations). We have modeled such correlations using exponentially decaying functions exp(-d/D) of the distances d between two observation pixels, and tested their inclusion in **R** with a 1-sigma uncertainty of 0.3 ppm for individual XCO2 data and with D=10, 50 or 100 km (assuming the correlations apply to the total uncertainty in individual data i.e. not splitting **R** into a random noise component and a component with spatial correlations). These tests are now described in Appendix A where the following Figure A3 is also provided. These results and this topic are now discussed in Section 4, with the following main messages:

- such isotropic correlations should make the errors much easier to filter by the inversion than actual spatial patterns of the systematic errors that can follow the same atmospheric dynamics as the signature of the targeted fluxes
- the situation is made even more optimistic here (like in most of studies with atmospheric inversion OSSEs) since our inversions are perfectly informed about the statistics of the observation errors (and in particular about their spatial scale)
- as expected, the results indicate that when introducing correlations with small spatial scales, the posterior uncertainties increase since these correlations yield larger budget of observation errors at the scale of the signatures of the targeted fluxes. Conversely, when introducing correlations helps the inversion separate the observation error patterns from these signatures, and large correlation spatial scales lead to a decrease of the posterior uncertainties. The worst case corresponds to correlation scales that are function of the scale of the signatures of the targeted fluxes. It varies depending on whether we analyze results for point sources, cities or widespread emissions across regions. This is why (i) results are poorer with D=10 km than with no correlations, for all type of sources (ii) results become generally better with spatial correlation than without for point sources and cities respectively for D≥10 km and D≥50km (iv) the inversion of the largest of a given category of sources (in terms of amplitude) are generally more negatively or less positively impacted by the spatial correlations (iii) results are systematically worse with D= 10 to 100 km than with no correlations, for countryside emissions. Results for regional budgets mix all these behaviors in a complex way.

These results and conclusions should be further investigated in future theoretical studies (to properly analyze the characterization of the spatial scale of the signature of the targeted fluxes) but this would fall out of the scope of this paper since (i) we want to keep the focus on the impact of random instrumental noise (ii) we believe that such a simple structure of the error correlations and their optimal learning by the inversion should be interpreted carefully to avoid misleading conclusions regarding the impact of actual systematic errors in the inversions.

Here is the corresponding text added in section 4:

*Our new inversion framework allows accounting for a realistic simulation of the observation sampling and errors. Nevertheless, generating simulations of the systematic errors from the retrieval of XCO$_2$ data that are suitable for the purpose of our study would have been difficult. Systematic errors are not described in the uncertainties computed by existing retrieval schemes. Furthermore, they depend on specific measurement configurations and on the evolving skill of radiative transfer inverse models and of empirical bias-correction*

*systems, so that their characterization based on diagnostics with existing missions may hardly apply to future ones. Simulating realistic patterns of cloud cover consistent with the meteorology for the different test cases investigated would have also been challenging. Finally, this study focuses on other parameters to allow exploring the sensitivity of the inversions to these parameters in depth.*

*In order to raise insights into the impact of errors with spatial patterns such as model and systematic errors, we have conducted experiments where spatial correlations are included in the observation error (in the $\mathbf{R}$ matrix). We have tested isotropic and homogeneous spatial correlations exponentially decaying with distance, using various correlation lengths. The experiments and results are described in Appendix A since they are out of the scope of this study. The results indicate that including correlations in the observation errors tends to increase the budget of observation errors and thus to increase the posterior uncertainties in the flux estimates as long as the correlation length scale does not exceed that of the signature of the fluxes in the $XCO_2$ images. However, including correlations in the observation errors also tends to increase the ability to distinguish between the patterns of the observation error and of the signatures of the fluxes, and thus to decrease the posterior uncertainties, so that for large spatial correlation lengths, increasing the correlation length leads to a decrease of the posterior uncertainties. In our tests, the worst situation for the monitoring of the emissions in the study area corresponds to ~10 km correlation length scales. These results should be interpreted cautiously since the spatial patterns of the model and systematic errors are more complex than this traditional but simple modeling of spatial correlations and since in these tests, the inversion system if perfectly informed about the statistics of the observation error. Future studies will integrate more realistic simulations of observation sampling and errors from different concepts of spaceborne imagery, based on radiative transfer inverse modeling applied to realistic fields of surface and atmospheric conditions and instrumental specifications*

[Figure]

*Figure A3 : Uncertainty reductions (UR) for the emissions of the point sources (1[st] row), the urban areas (2[nd] row), the countryside areas (3[rd] row) and the regions (4[th] row) when considering spatial correlations (Y-axis) or no correlations (X-axis) between the observation errors. The 1[st],2[nd] and 3[rd] column correspond to correlations with a spatial scale of 10, 50 and 100 km respectively. The color of the dots corresponds to the annual budgets of the sources (color bar on the right of the figure). The dashed line is the 1:1 line.*

and (ii) no spatial correlation in their prior flux covariance B (L334), also unrealistic given the high spatial resolution of their fluxes.

The characterization of the correlations of the uncertainties in anthropogenic emissions of CO2 from inventories (used as prior estimates) is an extremely complex topic, and, currently, there is certainly no consensus on how these correlations can be defined. In fact, one may argue for both positive and negative correlations. Positive spatial correlations arise from uncertainties in the emission factors applied at national scales in the inventories. Negative correlations arise from the spatial disaggregation of national inventories into gridded maps of emissions (Wang et al., 2018) use negative correlations between regions of the same country). Variations of activities from one city to another, from one plant to another… de-correlate

uncertainties. The use of local emission factors also de-correlates uncertainties. Finally, in a general way, the spatial correlations should be highly complex and have no reason to decrease with distance (as suggested by the reviewer's statement "given the high spatial resolution of their fluxes"), since e.g. uncertainties in emissions from two distant cities with large industrial activities should be more correlated than that between two neighbor cities with a completely different share of domestic, commercial, transport and industrial activities. State of the art estimates of the structures of uncertainties in the anthropogenic emissions (Super et al., 2020) strongly depend on the model used for the inventories and on the chosen assumptions regarding the major sources of uncertainties.

Given the current lack of characterization of correlations of the uncertainties in anthropogenic emissions of $CO_2$, ignoring them or assuming that they are null is a safe solution. We clarify this point in section 2.4 and in section 4:

*The spatial correlations of the uncertainties in anthropogenic emission inventories is a complex topic and the current lack of characterization for such correlations led to such a conservative set-up (Wang et al. 2018; Wang et al., 2020; Super et al., 2020).*

*There is still a critical lack of knowledge and of characterization of the correlations in the uncertainties in the inventories (Wang et al. 2018, Wang et al., 2020). However, some extensive analyses are now conducted to fill this gap (Super et al., 2020).*

NEE is controlled at regional scale only which implicitly translates into significant spatial correlations of the uncertainty in the bottom up estimates of the NEE.

I do not understand why they need to make either simplification, since in a batch inversion (as opposed to an iterative approximation like EnKF or 4DVAR) one can actually specify off-diagonals in both B and R. The simplifications (i) and (ii) make their posterior covariance, and the conclusions based thereon, not very relevant for real-world inversions of CO2M-like satellite data.

This is not about simplification but about making realistic scenarios as detailed above and we strongly disagree with the last statement of the reviewer. Of note is that, in a general way, adding spatial correlations in **B** and **R** tends to simplify the inversion problem, to spoil the asset of solving for the fluxes and concentration at high resolution, and to increase the scores of uncertainty reduction. We think it would lead to over-optimistic results.

A general comment I would like to make about the three issues I raise above is that in inversions of real satellite data, modelers often have to make simplifying choices to make the problem tractable. E.g., ignoring off-diagonal elements in R is pretty common, although more recent inversions try to at least account for correlations in R by error inflation, aggregating, or data thinning (e.g., https://doi.org/10.1029/2007GL030463, https://doi.org/10.5194/acp-13-8695-2013 and https://doi.org/10.5194/acp-19-9797-2019).

As detailed above, we can definitely include correlations in **R** and we did it in answer to both reviews. However, the corresponding set up and results are not relevant enough for our paper to be included within the result section so that they are kept for Appendix A.

Similarly, inversions with an iterative approximation like EnKF or 4DVAR often have an inexact posterior covariance due to computational limitations, while many inversions ignore biases in satellite retrievals because it's still an open problem. However, making too many simplifications in a single OSSE, as the authors have done here (unbiased priors and retrievals, no sampling gaps, uncorrelated retrieval errors, no spatial error correlation in fluxes) makes the results inapplicable to any real-world satellite instrument.

See our answers to all these general critics with which we obviously disagree.

I will also point out that several coauthors of this manuscript have previously authored papers stressing the importance of these complicating factors (e.g, https://doi.org/10.1029/2007GL030463, https://doi.org/10.5194/amt-11-681-2018, and https://doi.org/10.1088/1748-9326/ab7835) and published inversions with far more realistic assumptions, which makes the current set of simplifications all the more surprising.

We thank the reviewer for recalling these studies that highlighted various challenges which have often been disregarded in other studies. However, they should not be used as an obligation for us to solve all these long-standing challenges in this paper that is driven by its own scientific questions.

Chevallier (2007) studied the impact of hypothetical observation error correlations of 0.5 in neighboring observations supposedly caused by modelling errors (radiative transfer and atmospheric transport). Such errors are out of the scope of our study, as clearly stated in the submitted version, but, as we have explained, we have now included tests to touch this topic.

Broquet et al. (2018) explored the impact of errors in the emission spatial sampling and in the boundary conditions but discussed the shortcoming of testing these errors as biases due to the lack of ensemble simulations for these parameters. They studied the impact of cloud cover and systematic errors in the last step of their incremental analysis, after having diagnosed series of experiments ignoring it, in order to focus on other parameters first, which led to a major part of the conclusions. Our introduction and discussion / conclusion highlight the amount of new topics explored here in cloud free conditions without systematic errors.

The problem of the uncertainties in biofuel emissions and of human respiration from cities has been raised by Ciais et al. (2020) as a problem for the spaceborne observation but not tackled by any inversion system yet. Actually, this publication shows that this problem will have to be addressed by complementary data or source of information since one can hardly adapt an atmospheric inversion approach that assimilates satellite XCO2 images only. We now discuss this additional source of uncertainty in our results (section 4):

*A last significant simplification of the general problem of the inversion of the anthropogenic emissions based on $XCO_2$ data has been stressed by Ciais et al. (2020). Anthropogenic emissions of $CO_2$ bear a major share of emissions from biofuel combustion which can hardly be separated spatially from the fossil fuel combustion component. Furthermore, the emissions of $CO_2$ by human respiration represent a significant portion of the total $CO_2$ emitted from cities. The $XCO_2$ data and the atmospheric inversion approaches can hardly be used to distinguish between these different components if it cannot rely on complementary data. This factor was ignored here, as well as in most of the studies dedicated to the inversion of anthropogenic $CO_2$ emissions at city scale to regional scale.*

Beside these major issues, here are a few minor points that need correcting or clarifying:

1. L156: Delete "and vertical". The vertical resolution comes later.

Done

2. L161: A high spatial resolution (~2x2 km2) implies higher temporal resolution as well. If the driving winds are 3-hourly, what provides high frequency variation in the CHIMERE winds?

Before any simulation, a preprocessing stage interpolates the 3-houlry winds from ECMWF at 1h-resolution and at the spatial resolution of CHIMERE to generate the wind forcing in input to the model. CHIMERE itself interpolates the hourly forcing at each of its physical time step (Menut et al., 2013). To clarify this point, we add a sentence in section 2.1.1: *These three-hourly fields are interpolated at the spatial and temporal resolution of CHIMERE.*

3. L175: The gradients in column CO2 due to the top 30% of the atmosphere would be small, agreed, but how large are they? Signals in column CO2 are deceptively small, so terms that seem to be negligible are not always negligible.

The analysis of the $CO_2$ concentrations at 11:00 produced by the response functions associated to the emissions between 5:00 and 6:00 of several cities and point sources shows that these $CO_2$ concentrations at the upper level of the model are indeed negligible with respect to the ones within the boundary layer (with a factor < 1e-6). This supports our approximation to prescribe the $CO_2$ concentrations in the upper layers of the atmosphere. To clarify this point, we add in section 2.1.1 the sentence:
*This is supported by the lack of signal in our simulations of the atmospheric signatures of the surface fluxes in the upper layer of the model.*

4. L180: Switch 92.8° and 705 km.

Done

5. L227: Since the quantity directly estimated is the FF CO2 emission between 5 and 11 local time, to estimate the total emissions one would need an accurate diurnal cycle. What is the uncertainty in the diurnal cycle of FF CO2 emissions?

The structures of temporal correlations of the uncertainties in the estimates of emissions from inventories is a very complex topic (as well as that of their spatial correlations). The uncertainty in the diurnal cycle of the emissions has hardly been characterized and will strongly depend on the inventories that are considered. Some insights regarding these uncertainties can be found in Super et al. (2020).

However, this part of the paper was not addressing the topic of the estimate of the total emissions (over the day or over the year). Note that we have mentioned this topic at the end of section 4 (*Exploiting further capabilities of the inversion framework: potential of complementary observation systems and results at larger temporal scales*).

6. Section 2.2.2 and elsewhere: The word "controlled" keeps confusing me. Do you mean "estimated", as in part of the control vector? Or do you mean "controlled" as in kept in control, static, not changed? I suspect you mean the former, but "controlled" in English can also signify "not allowed to change". I'd suggest using the word "estimated" or "optimized" if you mean the former.

Throughout the text, we use the term "control – controlled (by the inversion)" in a unique, clear and explicit way that is equivalent to "correct / adjust / update - corrected / adjusted /

updated". Rayner et al. (2019) acknowledges this traditional denomination even though they promote the term « target »: « We term the set of these quantities the "target variables" of the problem. They are also called unknowns, parameters or control variables. » (they do not speak about « estimated » or « optimized » variables). We prefer « control » than « target » since the inversion often has to control some variables which are not a scientific target to explicitly account for and tackle uncertainties from these degrees of freedom. In this OSSEs framework where the posterior estimate of the control parameters is not derived, the term "estimated" and "optimized" would not be ideal, and we do not think that "estimated vector" or "estimated region" sounds very well. Finally, the term optimized can be misleading in statistical inversion, especially when focusing on the posterior error covariance matrices i.e. on the statistical distribution of the posterior uncertainty around its optimal estimate rather than on this optimal estimate.

7. L280: Is this an over-determined problem? Then that's not very common in flux inversions, and is likely due to the unrealistic correlations in R and B (one of my major concerns).

1) We are speaking here about the size of control and observation vectors in a statistical inversion problem, not about the question whether the problem is over or under constrained. Statistical inversion problems where the number of observations is larger than the number of control variables, especially when studying spaceborne imagery of XCO2, is quite common (e.g. Reuter et al., 2014; Kemp et al., 2014; Pillai et al., 2016).

2) Furthermore, adding correlations in **B** will implicitly reduce the size of the space of uncertainty in the fluxes, not the opposite.

3) Finally, mathematically speaking, changing correlations in **B** and **R** will not change the size of the control and observation vectors.

Regarding the major concerns on **R** and **B**, see the discussions above.

8. L290: Is random noise added to y, consistent with R?

At this stage of the paper, the text has not said that we will generate pseudo observation **y**. It just provides the theoretical basis for the following. The following clarifies the fact that we do not generate such pseudo-data.

9. L304: Typo, change XO2 à XCO2

Done

10. L348: With the assumptions detailed here at the grid scale, what is the uncertainty on the (say) annual total or seasonal total NEE and FF? Aggregate numbers are easier to make sense of than grid-scale specifications.

We do not want to enter into discussions about the uncertainties in annual fluxes and about their link with uncertainties at the 6-hour scale in this part of the paper. This requires discussing the temporal correlations of the prior uncertainties in inventories of the anthropogenic emissions at the daily to annual scale. Again, this is a highly complex topic (Wang et al., 2020; Super et al. 2020), and the general topic is discussed in the last part of

11. L399: "Figures 2i" likely means all the subplots of Figure 2. In that case, just say "Figures 2", no need to add the "i".

Done

12. L437: Speed is one aspect of the wind, direction is the other. Since wind direction determines how well plumes present themselves to a satellite that is going one way, uncertainty in wind direction must be considered as well as speed uncertainty. Was that done here?

This section does not discuss uncertainties in the wind but the sensitivity to wind variations. This specific part of the text demonstrates the influence of wind speed on the results. The impact of changes in the wind direction is significant for narrow swaths and discussed later.

13. L477: Remove "uncertainty", B is just the prior covariance matrix.

No, absolutely not. **B** is the prior uncertainty covariance matrix i.e. **B** is the covariance matrix of the uncertainty in the prior, not the covariance matrix of the prior.

14. L519: Again, I'd like to see the uncertainties on aggregated fluxes, such as annual totals.

We conduct 6-hour inversions. There is no need in this study to make assumptions on temporal correlations of the prior uncertainty on larger timescales to propose some corresponding annual uncertainties.

15. L560: In Figure 7, why do larger emissions have smaller uncertainty reductions?

The Figure 7 shows that the general tendency is to get larger uncertainty reductions for sources with larger emissions. However, other factors driving the variations in uncertainty reductions may soften this conclusion for sources that have similar emissions: e.g., if the plumes from sources are driven by highly different wind speeds. The influence of the wind on the inversion results is analyzed in sections 3.2.1 and 3.3.1.. And, as discussed in sections 3.2.2 and 3.3.1., the UR for a given source is also sensitive to the level of uncertainty in NEE around, and to the potential loss of part of its plume at the edge of the satellite swath. We clarify this point by adding the sentence in section 4: *Beside these sensitivities to the source amplitudes and to the uncertainties in NEE, the variations in UR are also driven by the wind (sections 3.2.1 and 3.3.1) and by the potential loss of part of the atmospheric signatures of sources at the edges of the satellite swath (section 3.2.2).*

16. L582: "… and thus by the variability of these fluxes, during the month of May". This only matters because the uncertainty on the NEE is larger in May, right? Because this metric/score does not care about the actual prior NEE.

Yes, it does, since we define relative values for the prior uncertainties in the NEE rather than absolute values for the prior uncertainties in the NEE. So if the NEE varies, the resulting absolute value of the corresponding prior uncertainty will vary too.
We clarify this point in section 3.3.1 with the sentence:

*Using constant prior relative uncertainties in the natural fluxes (as for the anthropogenic emissions) yields large absolute uncertainties in May and low absolute uncertainties in March.*

17. L794: "Efforts have been made to limit the amplitude of such errors in the concept of the new CO2M mission. Our new inversion framework allows accounting for a realistic simulation of the observation sampling and errors." I disagree with this statement in the context of this paper, especially the part about a realistic simulation, because of the three major points raised above.

Again, see our answers to these points. We do not think that this sentence has to be modified.

[revised manuscript text omitted]

---

## Author Response (AR2)

We thank the reviewer and the associate editor for their recommendations on our manuscript. In its new version, we have created a conclusion section and slightly lighten the discussion section. We have also highlighted in the abstract and in the conclusion that the OSSEs were performed with clear-sky conditions and that the inclusion of more realistic estimations of errors, especially systematic, may strongly impact the results.

We provide below the version of the new manuscript where the changes are highlighted.

[revised manuscript text omitted]